# Using a region-specific ice-nucleating particle parameterization improves the representation of Arctic clouds in a global climate model

Astrid B. Gjelsvik[1], Robert O. David[1], Tim Carlsen[1], Franziska Hellmuth[1], Stefan Hofer[1,a], Zachary McGraw[2,3], Harald Sodemann[4,5], and Trude Storelvmo[1,6]

[1]Department of Geosciences, University of Oslo, Norway
[2]Department of Applied Physics and Applied Mathematics, Columbia University, USA
[3]NASA Goddard Institute for Space Studies, USA
[4]Geophysical Institute, University of Bergen, Norway
[5]Bjerknes Centre for Climate Research, Bergen Norway
[6]Nord University Business School, Nord University, Norway
[a]Now at: School of Geographical Sciences, University of Bristol, UK

**Correspondence:** Astrid B. Gjelsvik (a.b.gjelsvik@geo.uio.no) and Robert O. David (r.o.david@geo.uio.no)

**Abstract.** Projections of global climate change and Arctic amplification are sensitive to the representation of low-level cloud phase in climate models. Ice-nucleating particles (INPs) are necessary for primary cloud ice formation at temperatures above approximately -38°C, and thus significantly affect cloud phase and cloud radiative effect. Due to their complex and insufficiently understood variability, INPs constitute an important modelling challenge, especially in remote regions with few observations, such as the Arctic. In this study, INP observations were carried out at Andenes, Norway in March 2021. These observations were used as a basis for an Arctic-specific and purely temperature-dependent INP parameterization, and implemented into the Norwegian Earth System Model. This implementation results in an annual average increase in cloud liquid water path (CLWP) of 70 % for the Arctic, and improves the representation of cloud phase compared to satellite observations. The change in CLWP in boreal autumn and winter is found to likely be the dominant contributor to the annual average increase in net surface cloud radiative effect of 2 Wm$^{-2}$. This large surface flux increase brings the simulation into better agreement with Arctic ground-based measurements. Despite that the model cannot respond fully to the INP parameterization change due to fixed sea surface temperatures, Arctic surface air temperature increases with 0.7°C in boreal autumn. These findings indicate that INPs could have a significant impact on Arctic climate, and that a region-specific INP parameterization can be a useful tool to improve cloud representation in the Arctic region.

## 1 Introduction

The Arctic has warmed almost four times more than the rest of the world since around 1980, due to anthropogenic climate change (Rantanen et al., 2022). The rapid warming has dramatic consequences for Arctic ecosystems, along with the livelihood of indigenous peoples and other Arctic communities. Amongst the global consequences are sea level rise due to glacier and ice sheet melt, and further temperature increase through loss of bright (high albedo) surfaces such as snow and sea ice. This

pronounced warming in the Arctic compared to the rest of the world is known as Arctic amplification (Serreze and Barry, 2011; Taylor et al., 2022). A number of different climate feedbacks have been proposed to explain it, including the surface albedo reduction through snow and sea ice loss, confinement of warming to the surface (lapse-rate feedback), increased poleward heat transport in the atmosphere and ocean, and cloud feedbacks (Forster et al., 2021). However, climate models have been shown to underestimate the present Arctic amplification (Rantanen et al., 2022; Hahn et al., 2021). The uncertainty in the models arise from uncertainty in the multitude of processes affecting Arctic amplification, including cloud feedbacks (Forster et al., 2021; Taylor et al., 2022), and their interaction with other processes (Hahn et al., 2021; Taylor et al., 2022).

The changing role of clouds in Earth's radiative budget with warming is arguably the largest uncertainty in determining the climate sensitivity of the Earth (Forster et al., 2021). Different cloud feedbacks can contribute to both amplifying and damping radiative forcings, by either trapping more (less) terrestrial radiation from the surface or reflecting less (more) incoming solar radiation. Part of the uncertainty is due to insufficient knowledge on climate feedbacks of cold clouds (Ceppi et al., 2016; Zelinka et al., 2020; Murray et al., 2021). These can consist of both ice and supercooled liquid water, in which case they are described as mixed-phase. In the mid- to high-latitudes, low-level clouds can contribute to a significant negative climate feedback as increasing temperatures will lead to larger fractions of liquid water in the clouds, which will likely increase the cloud albedo and lifetime (Forster et al., 2021). However, the magnitude of this negative feedback depends on the present supercooled liquid water (SLW) fraction in the clouds (Tan et al., 2016). How this fraction is represented in climate models is highly diverging, and leads to substantial differences in model climate sensitivity (Zelinka et al., 2020). Models with lower initial SLW fractions tend to favour a stronger Arctic amplification (Tan and Storelvmo, 2019; Zelinka et al., 2020), making the SLW fractions of mixed-phase clouds an important area of research for predicting the development of the Arctic and global climate.

In order to represent the SLW fraction correctly in climate models, it is important to correctly represent the concentration of the available ice-nucleating particles (INPs). These particles are necessary to initiate heterogeneous ice nucleation, occurring above approximately -38 °C (Kanji et al., 2017). The presence of INPs, therefore, plays an important role in modulating the concentration of ice crystals in cold clouds. However, the concentration of INPs is highly variable in space and time, and there is still insufficient knowledge of the sources and properties of these particles (Kanji et al., 2017). For the Arctic, marine organic aerosol particles from ocean biological activity have been presented as a potentially important source of INPs (DeMott et al., 2016; Wex et al., 2019; Creamean et al., 2019), in addition to mineral dust being transported from lower latitudes (Shi et al., 2022). Other relevant sources are local glacial dust, which Tobo et al. (2019) have found to have a potentially high ice-nucleating ability, and terrestrial vegetation (Pereira Freitas et al., 2023). More knowledge on INP concentrations is particularly important in the high-latitudes, where the low-level cold cloud feedback is especially relevant, but INP concentration measurements are few (Vergara-Ttemprado et al., 2017; Murray et al., 2021). With a rapidly changing Arctic climate, the sources of INPs are likely also undergoing rapid change, stressing the urgency in understanding their climate impact.

The objective of this study is to investigate how simulations of Arctic climate, specifically Arctic clouds and radiation, might change when applying INP concentrations constrained by region-specific observations. We have conducted field measurements of INPs in Andenes on the Norwegian island Andøy, located north of the Arctic circle. To do this, we used the newly developed

DRoplet Ice Nuclei Counter Oslo (DRINCO) (based on the instrument DRINCZ David et al. (2019) and FINC Miller et al. (2021)) to quantify INP concentrations from collected air samples. The measurements were conducted in March of 2021, a period of frequent outbreaks of polar air masses (cold air outbreaks) reaching the measurement site through northerly winds, similar to the COMBLE campaign in 2020 (Geerts et al., 2022). The observations from Andenes in March 2021 form the basis of a new parameterization for Arctic INP concentrations, which is implemented into the second generation of the Norwegian

Earth System Model (NorESM2) (Seland et al., 2020b).

Previous modelling studies have included parameterizations for marine organic aerosols to better represent INP concentrations in marine environments such as the Arctic in global climate models. The first was implemented by Yun and Penner (2013), and later modelling attempts have shown varying importance of marine organic aerosols for INPs (Huang et al., 2018; Vergara-Temprado et al., 2017; McCluskey et al., 2019). The recent parameterization of Zhao et al. (2021) shows promising

ability to reproduce INP concentrations, but more work is still needed. Ours is a more simplified parameterization based only on temperature, but is tailored specifically for the Arctic region, and is restricted to latitudes above 66.5°N. In this sense, we are complementing work that has been done previously by English et al. (2014), and the Arctic-specific dust parameterizations for global models of Shi et al. (2022) and Kawai et al. (2023). Our purpose with observationally constraining Arctic INPs in a simplified manner is to limit bias sources in the simulation, and investigate the potential of simple aerosol-independent

parameterizations when computational resources are a limiting factor. Building on the master's thesis Gjelsvik (2022), we will demonstrate that such an approach has a substantial effect on Arctic clouds and radiation, and leads to improved representation of Arctic clouds compared to observations.

## 2  Methods

### 2.1  Aerosol sampling

The field measurements presented here were conducted at Andøya Space (AS) in Andenes, Norway (69°18' N, 16°07' E) from 2021-03-15 to 2021-03-30. The measurements were conducted as part of a joint campaign between the University of Oslo and the University of Bergen.

The aerosol sampling site (marked with a red dot in Fig. 1b) was situated at sea level around two hundred meters from the North-facing shore, and is shielded from the south by mountains that rise approximately 200-400 meters above sea level.

The ambient aerosols were sampled through a 6-meter-high home-built aerosol inlet. The inlet was heated ($\sim 16°C$) to ensure that all cloud particles were completely evaporated/sublimated before entering the instruments and prevent the build-up of snow and rime from restricting the airflow through the inlet. At the base of the inlet, the flow temperature was monitored (Type K thermocouple recorded by an EL-GFX-TC, Lascar Electronics datalogger) in order to report the aerosol and INP concentrations per standard litre. For sampling, the flow was split to an optical particle counter (OPC; Met One GT-526S) and

a three-way ball valve (Model 120VKD025-L, Pfeiffer Vacuum, Germany), which in turn was connected to a high flow-rate liquid impinger (Coriolis-$\mu$, Bertin Instruments, France) and a blower (Model U71HL, Micronel AG, Switzerland). When the

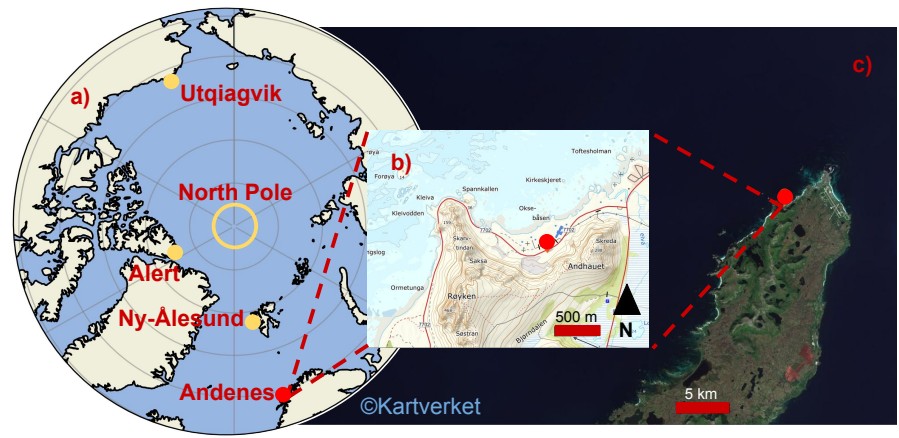

**Figure 1.** Location of INP measurement site. In panel (a), the position of the measurement site at Andenes, Norway, is shown with a red dot, together with the other Arctic study regions Alert, Ny-Ålesund, Utqiagvik and the North Pole (marked in yellow). In panel (b), the measurement site is marked in the immediate surrounding area, while in panel (c) the site location on Andøy is shown, both with red dots. The maps in (b) and (c) are from ©norgeskart.no (Kartverket, 2022).

Coriolis-$\mu$ (described in Sect. 2.1.2) was not sampling, the ball-valve was rotated such that the blower maintained the 300 Lmin$^{-1}$ flow through the inlet, as described in Li et al. (2022).

Out of 52 samples in total, 51 are included in this study, excluding one sample measured on 2021-03-19 during a period of substantial wave breaking on the nearby ocean surface, which likely led to sea spray entering the aerosol inlet directly. This sample has estimated INP concentrations, which were clear outliers compared to other samples, and was therefore excluded from the parameterization of Arctic INP concentrations.

### 2.1.1 Aerosol properties

The OPC measured the number of particles per litre of air exceeding certain sizes in bins, i.e. the number of particles with diameter greater than 0.3 µm, 0.5 µm, 0.7 µm, 1 µm, 2 µm and 3 µm. We restrict our analysis to aerosols with diameters of 0.5 µm or larger when comparing aerosol and INP concentrations as this is the cutoff size of the Coriolis-$\mu$, as well as the size that has traditionally been considered most relevant for INPs (DeMott et al., 2010; Kanji et al., 2017). We calculate the aerosol surface area assuming a spherical particle shape, and the same size for all particles $\geq 3$ µm.

### 2.1.2 Coriolis setup

The Coriolis-$\mu$ collected aerosols for 40 minutes with a flow rate of 300 Lmin$^{-1}$ for a total of 12 m$^3$ of air per sample for subsequent INP analysis. As the air is sampled by the Coriolis-$\mu$, it rotates inside a cone-shaped flask causing particles larger

than $\sim 500$ nm (aerodynamic diameter) to be scavenged by purified water (W4502, Sigma-Aldrich, USA) due to their inertia. As some of the water evaporates during sampling, additional purified water is pumped into the flask at a fixed rate (e.g. 0.4 mL/min) during sampling. At the end of the sampling period, the resulting water volume of the cone is measured, and the ice-nucleating ability of the particles immersed in the purified water is prepared for further analysis.

### 2.1.3 INP analysis

The ice-nucleating ability of the collected aerosols was quantified using the DRINCO. DRINCO is based on the DRoplet Ice Nuclei Counter Zurich developed by David et al. (2019), with some updates described in Miller et al. (2021). DRINCO consists of an ethanol chilled bath (Julabo, FP51), a custom PCR tray holder, an LED light array, which is placed in the bath, a webcam (ELP-USB8MP02G-SFV, SVPRO, China) and a bath leveller composed of an optical level sensor (LLC102000, SST, UK) and a peristaltic pump (KAS-S10, Kamoer, China). During an experiment, the temperature of the ethanol bath is cooled at a constant rate of $1°$C/min and the webcam captures the light emission through a 96-well PCR tray (732-2386, VWR, USA) partially submerged in the bath every $0.25°$C. Each well of the 96-well PCR tray is filled with a 50 $\mu$L aliquot of sample ($V_a$) and the webcam captures the freezing of each aliquot as a decrease in the light intensity due to the lower light transmission through ice relative to water. This results in a frozen fraction (FF) of wells as a function of temperature (FF(T)). To convert this FF to a meaningful INP concentration as a function of temperature (INP(T)), the formulation by Vali (1971) is used. From Vali (1971) the cumulative INP concentration can be calculated as follows:

$$\text{INP}(T) = \frac{-\ln(1 - \text{FF}(T))}{V_a C_{\text{air}}}, \tag{1}$$

where, $V_a$ is the aliquot volume in the PCR tray. The conversion factor $C_{\text{air}}$, following Li et al. (2022), converts this number to the estimated ambient INP concentration in the ambient air as

$$C_{\text{air}} = \frac{V_{\text{air}}}{V_{\text{sample}}} \frac{p}{1013.25\text{hPa}} \frac{273.15\text{K}}{T} \tag{2}$$

where $V_{\text{air}}$ is the volume of air sampled by the Coriolis-$\mu$ ($\sim 12 \text{ m}^3$), normalized to standard litre using the air flow temperature measurements ($T$) and the ambient sea level pressure ($p$) retrieved from MET Norway. $V_{\text{sample}}$ is the volume of water in the Coriolis-$\mu$ cone at the end of each sample period. The temperature differences between the stated bath temperature and the actual temperature across the PCR tray are accounted for by recording the temperature in the wells in a run using ethanol instead of water in each well, and applying the average temperature offset as a calibration factor. The uncertainty of the instrument is estimated to be $0.9°$C (David et al., 2019).

## 2.2 Modelling with NorESM2

### 2.2.1 Model description

NorESM2 (Bentsen et al., 2013; Iversen et al., 2013; Kirkevåg et al., 2013, 2018; Seland et al., 2020b) is based on the second generation of the Community Earth System Model (CESM2, Hurrell et al., 2013; Danabasoglu et al., 2020). The two models

share code infrastructure and many of the same characteristics. As the goal of this modelling study is to investigate the impact of INPs on clouds and radiation, only the atmospheric component of NorESM2 (CAM6-Nor) is used (Seland et al., 2020a). CAM6-Nor differs from the Community Atmosphere Model of CESM2 (CAM6) in its use of a different atmospheric aerosol module (OsloAero6, Kirkevåg et al., 2018). The module also differs from CAM6 in its improved conservation of energy and momentum, as well as its parameterization of turbulent air-sea fluxes (Toniazzo et al., 2020). Stratiform cloud microphysics are handled by a two-moment scheme from Morrison and Gettelman (2008). Both CESM2 and NorESM2 have been used to contribute to the sixth and latest generation of the Coupled Model Intercomparison Project (CMIP6, Eyring et al., 2016).

### 2.2.2 Adjustments to cloud ice production in CAM6-Nor

CAM6 and CAM6-Nor both use a heterogeneous ice nucleation scheme based on classical nucleation theory (CNT), following Hoose et al. (2010). As the primary ice production scheme in CAM6 has a well-documented bug, consisting of an ice number concentration limit that has been shown to prevent heterogeneous nucleation processes from nucleating ice crystals (Shaw et al., 2022), it is unsuitable for studying sensitivities to INP concentration adjustments. Additionally, the CNT scheme calculates ice nucleation based on temperature and the surface area of dust and black carbon aerosols in each time step. However, studies have questioned the relevance of black carbon as an INP (Vergara-Temprado et al., 2018; Kanji et al., 2020; Schill et al., 2020), and marine organic aerosols might be of equal or more importance than dust for the Arctic (Creamean et al., 2019; Wex et al., 2019; Carlsen and David, 2022). Thus, a purely dust and BC-based INP scheme is, either way, less suited for our purposes. Based on these two factors, in combination with the fact that we fail to observe a relationship between INP concentrations and total surface area of aerosols with diameter $\geq 0.5\,\mu m$ at Andenes (see Fig. A1), we revert to the heterogeneous ice nucleation parameterizations of CAM5.

In CAM5, the different heterogeneous ice nucleation pathways in mixed-phase clouds are parameterized independently, namely, contact freezing (Young, 1974), immersion freezing (Bigg, 1953) and deposition and condensation freezing (Meyers et al., 1992). Here, we update the Meyers et al. (1992) parameterization (hereafter: "M92") in the Arctic, using our measurements from Andenes. This parameterization is active in the temperature range -37°C to 0°C, and is responsible for more than 90 % of ice crystals formed in CAM5 mixed-phase clouds (English et al., 2014). Since the measured INP concentrations are relevant for the immersion mode, replacing the M92 parameterization with our measurements entails excluding deposition and condensation freezing in Arctic mixed-phase clouds. This exclusion is justified by observational studies that found deposition and condensation freezing to be negligible for mixed-phase clouds (Ansmann et al., 2009; Boer et al., 2011; Westbrook and Illingworth, 2011). However, our exclusion of deposition freezing does not apply to temperatures below -37°C (cirrus regime). As we update the M92 parameterization using our INP measurements in immersion freezing mode, we exclude the Bigg (1953) immersion freezing parameterization. This is done without changing the routine for heterogeneous freezing of rain drops, which also follows from Bigg (1953).

In addition to changing the heterogeneous ice nucleation scheme in CAM6-Nor, we remove the ice number limit for Arctic mixed-phase clouds. To compensate for potential large increase in ice number due to the removal of the ice number limit, secondary ice production is limited to $1000\,\mathrm{m^{-3}s^{-1}}$, following Shaw et al. (2022). Rime-splintering mechanism is the only sec-

ondary ice production mechanism in the model, active at temperatures between -8°C and -3°C. Additionally, the detrainment of cloud particles through convective updrafts is shifted from having the ice phase fraction of detrained particles decreasing linearly with temperature, to having all cloud particles detrained as liquid above -35°C, following Hofer et al. (2024).

### 2.2.3 Model experiments

The model experiments conducted in this study are referred to as "M92" and "A21". M92 is a CAM6-Nor set-up, following the ice production adjustments described above, but using the standard CAM5 heterogeneous ice nucleation schemes. In A21, the activated ice number produced by the parameterizations of Meyers et al. (1992) and Bigg (1953) is replaced if the latitude exceeds 66.5°N. For these latitudes, the immersion freezing INPs in the temperature range -37°C to 0°C, are determined by the INP observations at Andenes instead. The new parameterization is a temperature-dependent exponential fit of the Andenes INP

measurements (see Fig. 2). For both experiments, we use the atmosphere component with a $2.5° \times 1.875°$ ($\sim 2°$) horizontal resolution, 32 hybrid-pressure layers in the vertical and a "rigid" lid at 3.6 hPa (40 km) (Seland et al., 2020b). To reduce the required simulation time for the differences due to the different heterogeneous ice nucleation parameterizations to emerge, both simulations are nudged to ERA-interim reanalysis data (Dee et al., 2011) using pressure and wind fields. The nudging still allows for the atmospheric component of the model to respond to changes in forcing due to the different parameterizations

used, while having the added benefit of removing climatic variability between the simulations (Dee et al., 2011). As such, we do not investigate the significance of differences between the simulations, as all changes are expected to be a direct result of the changes in the representation of heterogeneous ice nucleation. It is important to note, that as the other components of the model (e.g. ocean, land) are represented using input files, only the simulated temperatures over land and sea ice can respond to energy fluxes from the atmosphere, while sea surface temperatures are fixed to an observed 10-year climatology around

the year 2000. The simulations presented here are three years long with an additional three months for model spin up time. The specific simulation period, excluding spin up, is therefore 2007-04-01 to 2010-03-31. The calculation of modelled cloud radiative effect at surface and the top-of-the-atmosphere (TOA), as well as estimated cloud longwave emissivity, is shown in Sect. B2 and Sect. B3, respectively.

### 2.3 Cloud phase metrics and CALIOP lidar comparison

In order to compare the output of our model experiments directly with lidar observations, we use the same method as Shaw et al. (2022) to generate output of SLW fraction for two cloud phase metrics, one for "cloud top" and one for "cloud bulk". These are generated by filtering the overlying cloud optical thickness (COT), first by discarding the uppermost layers with COT < 0.3 to avoid including cirrus clouds and then selecting the highest layer of clouds within the mixed-phase temperature regime, which is categorised as "cloud top". Second, the "cloud bulk" is acquired by selecting all cloud layers with 0.3 < COT

< 3.0. The lidar observations are from NASA's Cloud-Aerosol lidar with Orthogonal Polarization (CALIOP, Winker et al., 2009). The instrument can discriminate spherical water droplets from non-spherical ice crystals in clouds by the ratio of the perpendicular and parallel polarization of the backscattered light from the lidar beam. We use CALIOP data averaged over the observational period 2009-06-01−2013-05-31. The observations are binned down to a $1° \times 1°$ resolution for comparison with

the model output. The SLW fraction is calculated on isotherms from -40°C to 0°C, with a 5°C increment. The observed SLW fraction is calculated as the the ratio of the number of liquid cloud top pixels to the sum of ice plus liquid cloud top pixels, following Bruno et al. (2021). The modelled SLW fraction is calculated as the ratio of cloud liquid surface area density to the sum of liquid and ice surface area densities, based on the method of Tan et al. (2016), which instead used cloud ice and cloud liquid mixing ratios.

Furthermore, we use observations of cloud fraction, ice cloud fraction and liquid fraction to evaluate the model simulations. In order to make a direct comparison, we use the GCM-Oriented Cloud-Aerosol Lidar and Infrared Pathfinder Satellite Observation (CALIPSO) Cloud Product (GOCCP, Chepfer et al., 2010), and compare it to the model output using the CFMIP Observation Simulator Package (COSP, Bodas-Salcedo et al., 2011). It should be noted that spatial averaging is required to make a direct comparison between satellite observations and the model, and that different cloud climatologies from remote sensing vary significantly between themselves, making CALIPSO-GOCCP no absolute truth (Chepfer et al., 2010).

In both cases, the data is spatially averaged over the Arctic, for latitudes above 66.5° and up to 82°, which is the northernmost limit of the lidar observations.

## 2.4 Radiative flux observations

The modelled radiative fluxes are compared with observations both for the top-of-the-atmosphere (TOA) fluxes and the surface fluxes in the Arctic. We retrieve TOA fluxes from the Clouds and Earth's Radiant Energy Systems (CERES) Energy Balanced and Filled (EBAF) data product (Loeb et al., 2018), and corresponding uncertainty estimates from NASA (2021). Surface fluxes are retrieved from three different Arctic measurement stations in the Baseline Surface Radiation Network (BSRN); Alert (82°30'N, 62°22'W), Utqiaġvik (71°17'N, 156°47'W) and Ny-Ålesund (78°55'N, 11°56'E). The location of the stations can be found in Fig. 1, and radiation data is provided by Cox and Halliwell (2021), Riihimaki et al. (2023) and Maturilli (2020), respectively. The uncertainty standards for the BSRN data is given by McArthur (2005).

## 3 Results and discussion

### 3.1 Observed ice-nucleating particle concentrations at Andenes

The measured ambient INP concentrations spanned two orders of magnitude at -15°C, and ranged between $10^{-4}$ and $10^{-1}$ INP L$^{-1}$ within the temperature range investigated (see Fig. 2). These values are consistent with recent INP observations conducted in Ny-Ålesund (78°55' N, 11°56' E) during autumn 2019 and spring 2020 by Li et al. (2022) and in Nordmela, NO (69°8' N, 15°40' E) located 25 km south of Andenes during the winter and spring of 2020 (Geerts et al., 2022). In Fig. 2, the measurements are shown together with an exponential fit of the data, as well as the corresponding Ny-Ålesund fit from Li et al. (2022) and parameterizations of seasonal INP observations in northern Greenland (Sze et al., 2023). The INP measurements from Nordmela (Geerts et al., 2022) are also shown as grey crosses in Fig. 2, and exhibit a similar variability as the Andenes measurements, except for a lower concentration of INPs at freezing temperatures above -15°C and a more

extensive measurement temperature range. The more remote Greenland site has lower winter INP concentrations than Andenes in spring, while Greenland in summer has slightly larger concentrations of high-temperature INPs. The seasonal Arctic INP cycle has often been related to local biological marine or terrestrial sources (Creamean et al., 2022; Carlsen and David, 2022; Sze et al., 2023; Pereira Freitas et al., 2023) with higher activity in summer. Arctic glacial dust with high ice-nucleating ability has also been shown to play an important role in the lower troposphere during summer months (Tobo et al., 2019; Kawai et al.,

2023; Tobo et al., 2024), in addition to the variability coming from low latitude dust sources throughout the year (Shi et al., 2022). It should be noted that our measurements are a snapshot in time. The seasonal Arctic INP cycle certainly contains both lower and higher INP concentrations compared to our measurements, both related to the aforementioned local sources as well as transport from lower latitudes. Importantly, our measurements are only at the surface level, while studies from Shi et al. (2022) and Raif et al. (2024) show that long-range transported dust can play an important role at higher altitudes.

The proximity to year-round sea-ice free ocean at both Andenes and Ny-Ålesund could explain why INP concentrations in early spring are higher there than winter concentrations in Greenland. Following the trajectories of the air parcels reaching the boundary layer at the Andenes site during measurement times (see Fig. A2), we see that they largely pass through either the Greenland Sea or further south in the North Atlantic in the time prior to the INP measurements. The INP concentrations we measure therefore seem to indeed mainly come from air masses travelling over open ocean, both associated with polar air

travelling south (so-called cold air outbreaks), and with southerly winds going into the Arctic. This contrasts them with the measurements from Nordmela, which were targeted towards cold air outbreaks (Geerts et al., 2022). While our measurements are not exclusively from air with Arctic origin, measuring INP concentrations in air going into the Arctic is also relevant to Arctic INP concentrations.

     When comparing the various Arctic INP parameterizations to the parameterization of Meyers et al. (1992), it is immediately

evident that the M92 fit is not representative of the typical INP concentrations in the Arctic (see Fig. 2), as found in earlier work by e.g. Prenni et al. (2007). This is also not expected, as the M92 fit was based on INP measurements in Wyoming (Rogers, 1982) and Manchester (Al-Naimi and Saunders, 1985), far from – and likely not representative of – the Arctic. The M92 fit has almost two orders of magnitude higher INP concentrations than our measurements at the coldest temperatures and four orders of magnitude higher INP concentrations at the warmest temperatures.

We find that the ambient concentration of aerosols $\geq 0.5$ μm does not explain the variability of the 50 % FF temperature (Fig. A1d; $R = 0.25; R^2 = 0.06$). This is contrary to previous studies, e.g. by DeMott et al. (2010), but is consistent with recent Arctic INP measurements (Li et al., 2022). As noted, total surface area of aerosols with diameter $\geq 0.5$ μm and INP freezing temperatures show a low correlation as well ($R = 0.28; R^2 = 0.08$), which can be seen in Fig. A1e. This lack of correlation suggests that our INP measurements are dependent on a subset of particles with diameter $\geq 0.5$ μm, the minimum cutoff size

of our INP collection process, that do not covary with the overall concentration of larger particles. Explaining the variability of the INPs requires more knowledge of the particle composition than we have from our measurements.

     In the following sections (Sect. 3.2.1 and Sect. 3.3.1), we investigate the impact on cloud phase and radiation from replacing the M92 parameterization in NorESM2 with the observationally-based Andenes 2021 parametrization (hereafter: A21) in the Arctic (latitudes $\geq 66.5$°N). Due to the lack of correlation with aerosol $\geq 0.5$ μm, we implement the parameterization with

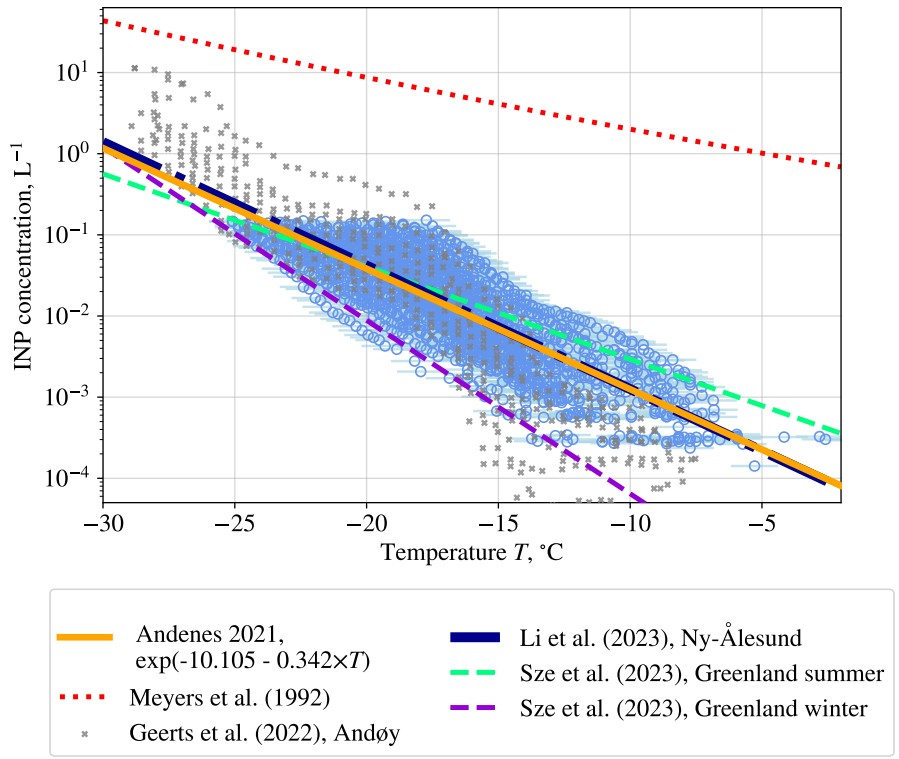

**Figure 2.** Ice-nucleating particle (INP) concentrations measured from the air at Andenes between 2021-03-15 and 2021-03-30, shown with blue circles. The light blue error bars show the uncertainty of the measurements. The orange line is the parameterization of INP concentrations as a function of temperature ($R^2 = 0.59$). For comparison, the INP study of Li et al. (2022) from Ny-Ålesund (dark blue), the Greenland summer (green) and winter (purple) study by Sze et al. (2023) is included, as well as the parameterization of Meyers et al. (1992) used in NorESM2. The grey crosses mark INP measurements during cold air outbreaks at Andøya in 2020, as part of the COMBLE campaign in Nordmela (Geerts et al., 2022). The values are taken from visual inspection of Fig. 7 in Geerts et al. (2022).

temperature dependence only, using the exponential fit seen in Fig. 2 to represent the number of activated INPs in mixed-phase clouds as a function of temperature (see Sect. 2.2.3).

### 3.2  Modelled cloud phase and comparison with observations

#### 3.2.1  Modelled cloud phase

Figure 3 shows the effect of replacing the M92 parameterization with the observationally-based A21 parameterization on ice
cloud fractional occurrence (Fig. 3a), and modelled grid-box averaged ice number concentration (Fig. 3b and 3c). Ice clouds are defined here as clouds with an ice mixing ratio larger than $10^{-6}$, and can also contain supercooled liquid water. First and foremost, there is a large decrease of around 0.05 in the ice cloud fractional occurrence for the Arctic as a whole, up to heights

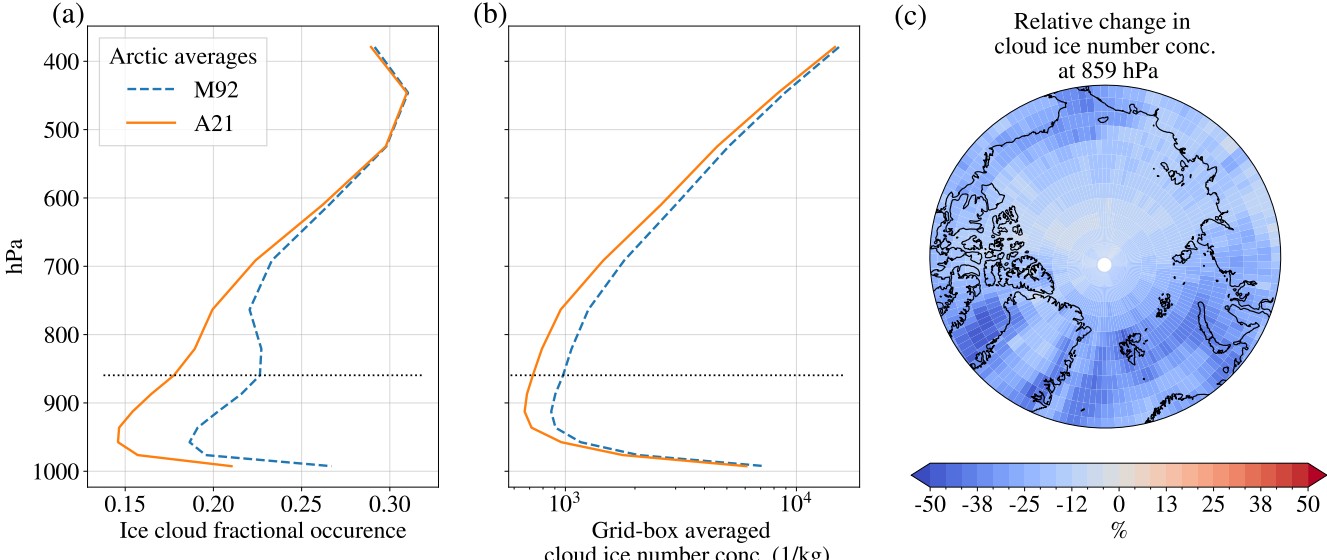

**Figure 3.** The average ice cloud fractional occurrence (a) and grid-box averaged cloud ice number concentration (b) for M92 and A21, averaged over the period 2007-04-01 to 2010-03-31. The profiles are averages over all latitudes above 66.5°N for height levels in hybrid sigma pressure coordinates (midpoint). The height level is marked with a black dotted line in panel (a) and (b). The relative change in ice cloud fractional occurrence from M92 to A21 at pressure level 859 hPa is shown in (c). Areas with open ocean area $\geq 85\%$ for more than 50 % of the year are hatched with black dots in (c).

around 800 hPa (see Fig. 3a). One of the largest reductions in ice cloud fraction is found at the 859 hPa level, where a local ice cloud maxima in M92 has nearly disappeared in A21 (see Fig. 3a). This change comes from a decrease in ice crystal number, which can be explored by looking into the grid-box averaged cloud ice number concentration. The reason we consider grid-box averaged values, that also include averages over ice cloud free areas, is because the pronounced reduction in ice clouds seen in Fig. 3a implies two very different ice cloud populations between the model experiments. Many of the ice clouds in M92 have been entirely transformed to liquid clouds in A21, which makes it less straightforward to compare in-ice-cloud quantities with M92. In Fig. 3b, we see a decrease in the cloud ice number of around 200 kg$^{-1}$ at height levels between 900 and 750 hPa. If we consider annually averaged relative ice number changes at 859 hPa, where we had the largest ice cloud fraction decrease, we see that there is a large reduction over the Arctic in general, of up to 50 % in some places (see Fig. 3c). There are slightly stronger reductions over places that are in closer proximity to open ocean and warmer surface temperatures, most prominently the Bering strait, Baffin Bay and the Norwegian Sea. While perhaps being more susceptible to INP parameterization changes in the first place, due to warmer temperatures, these areas are generally more cloudy as well, making it easier to notice changes in the grid-box averaged values.

The total grid-box cloud ice water path (CIWP) and cloud liquid water path (CLWP) change, separated by season, are seen in Fig. 4 and Fig. 5, respectively. Here, the changes are shown in absolute numbers in order to compare more directly with the

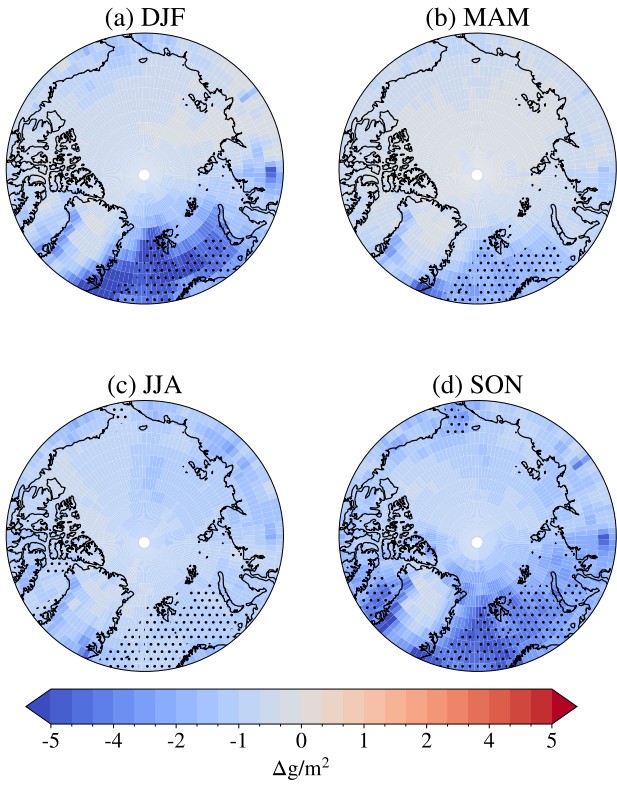

(a) DJF     (b) MAM

(c) JJA     (d) SON

-5  -4  -2  -1  0  1  2  4  5

$\Delta g/m^2$

**Figure 4.** Differences in total grid-box cloud ice water path between A21 and M92 by season (averaged by season over the period 2007-04-01 to 2010-03-31). Negative values correspond to lower cloud ice water path in A21 compared to M92. Areas with open ocean area $\geq 85\%$ for more than 50 % of the season are hatched with black dots.

absolute changes in cloud radiative effect in Sect. 3.3.1. The relative changes for selected areas in CIWP (CLWP) can be found in Fig. 6b (Fig. 6d).

The strongest decrease in CIWP happens over the Norwegian Sea in boreal autumn (SON) and winter (DJF), with some smaller changes in spring (MAM, see Fig. 4). In summer (JJA), the decrease is quite uniform over the whole Arctic, but of a slightly lower magnitude. For the Arctic in general, the largest decrease in CIWP appears in autumn, as can also be seen from the blue line in Fig. 6a. The average all-year relative change in the whole Arctic is around -14 %.

The CLWP changes (see Fig. 5) follow a similar spatial pattern as the CIWP changes, except that a decrease in CIWP

corresponds to an increase in CLWP, with a much larger magnitude. For example, the decrease of around 4 gm$^{-2}$ in CIWP that we see over Ny-Ålesund in October in Fig. 6a corresponds to an increase of 40 gm$^{-2}$ in CLWP in Fig. 6c. The average all-year relative increase in CLWP over the Arctic is around 70 %.

This large difference in magnitude between CIWP and CLWP changes could be explained by the Wegener–Bergeron–Findeisen (WBF) process (Wegener, 1911; Bergeron, 1928; Findeisen, 1938; Storelvmo and Tan, 2015). The rapid growth of ice crystals

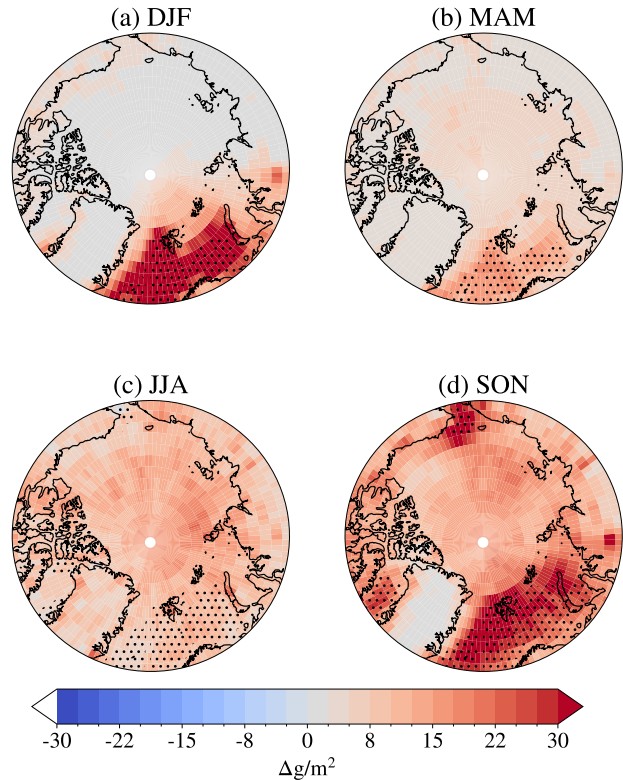

**Figure 5.** Differences in total grid-box cloud liquid water path between A21 and M92 by season (averaged by season over the period 2007-04-01 to 2010-03-31). Positive values correspond to a higher cloud liquid water path in A21 compared to M92. Note that the colorbar extends to sixfold the extent of the colorbar in Fig. 4. Areas with open ocean area $\geq 85\%$ for more than 50 % of the season is hatched with black dots.

can cause the clouds to dissipate faster, as the particles reach large enough sizes to fall out from the cloud at a quicker rate than through the growth of liquid particles. A significant reduction in ice particles can contribute to a reduction in the efficiency of the WBF process, and thereby lead to a liquid water content in the clouds much higher than the reduction in ice content itself, as the liquid water is no longer changing phase and precipitating out of the cloud at the same rate. Indeed, where we see large increases in total cloud water path, particularly in boreal autumn and winter (see Fig. B3) we also see reductions in total

precipitation (see Fig. B4), keeping in mind that such changes require careful interpretation as long as sea surface temperatures and evaporation is prescribed. Interestingly, we see that while the relative change in CIWP was largest over Ny-Ålesund in Fig. 6a, the largest relative changes in CLWP are over Alert and the North Pole, where we see extremely large relative changes, reaching over a 1000 % change or more in some seasons. These are places with very low CLWP in M92, causing the change in CLWP in A21 to make a relatively larger impact than over Ny-Ålesund, even though the absolute change is much higher here.

The annual mean total cloud water path north of 60 °N is 53.0 $\text{gm}^{-2}$ in A21 and 48.0 $\text{gm}^{-2}$ in M92. Both of these values are on the lower end of the range in mean cloud water path (49.5 $\text{gm}^{-2}$ to 82.7 $\text{gm}^{-2}$) estimated by reanalysis products in this

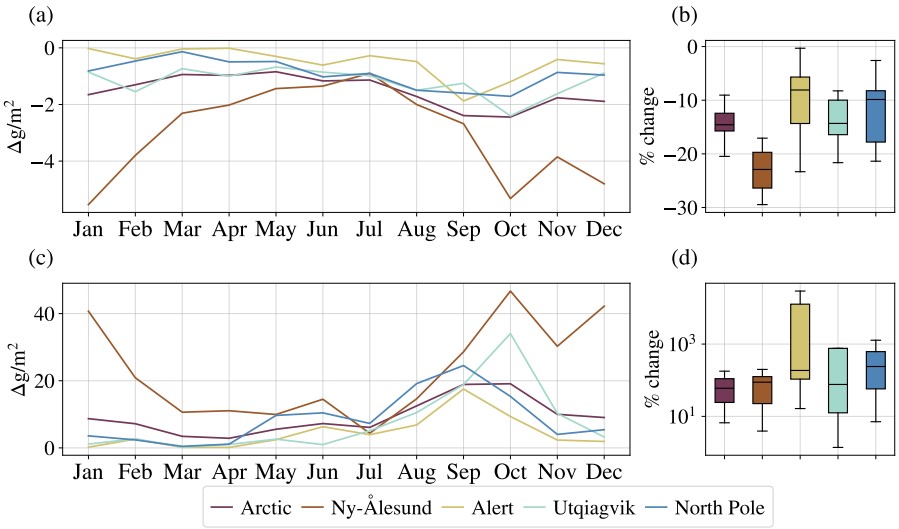

**Figure 6.** The change in total grid-box cloud ice water path (a-b) and liquid water path (c-d) between A21 and M92, averaged over the period 2007-04-01 to 2010-03-31. Left: absolute change for each month averaged over selected regions. Right: the distribution of relative change over the months in the same regions.

region (Gu et al., 2021). It should be noted that measurements of total cloud water path are known to have a large uncertainty at high latitudes (Khanal et al., 2020), particularly due to difficulties in separating precipitation and cloud particles.

### 3.2.2 Comparison to observed supercooled liquid water fractions

In order to see how the changes in modelled cloud phase due to the parameterization adjustment compares to actual observations, a comparison between the SLW fraction for each cloud isotherm in the model and in CALIPSO lidar observations is included in Fig. 7.

For the bulk of the cloud (dashed lines), the A21 parameterization adjustment produces SLW fractions much closer to the observations than M92. M92 shows around 20 % lower SLW fraction for temperatures between -25°C and -10°C. A21
reduces this gap to around 5 % less SLW fraction compared to observations, with virtually all A21 SLW fraction values falling within one standard deviation of the lidar measurements. For cloud top, the A21 experiment overestimates the SLW fractions, especially for temperatures less than -15°C. At colder temperatures of -25°C and -30°C, M92 and A21 diverge substantially for cloud top, with a difference in SLW fraction as large as 60 %. While it is clearly seen that the unrealistically high Arctic INP concentration in M92 causes excessive ice formation, it is less clear why A21 performs well for the bulk of the cloud,
while overestimating somewhat the cloud top SLW fraction. The fact that secondary ice production is limited to $1000 \, \mathrm{m}^{-3}\mathrm{s}^{1}$ is not likely to play a role in overestimating SLW, as the parameterization is only active at temperatures between -8°C and -3°C. At these temperatures, the cloud top SLW fraction in both model experiments and observations is close to 100 %. However, the fact that our model setup hinders ice detrainment at mixed-phase temperatures could perhaps be cutting off a relevant source of

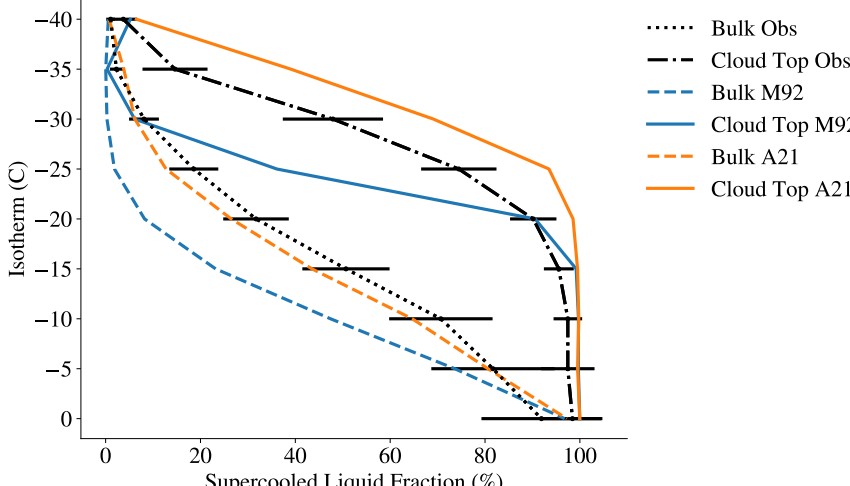

**Figure 7.** The supercooled liquid water (SLW) fraction for each isotherm in clouds for latitudes above 66.6°N and below 82°N. The dashed lines show the fraction for bulk cloud in the modelled climate, and the solid lines show the fraction for cloud top (CT) in the modelled climate, both in M92 (blue) and A21 (orange), averaged over the period 2007-04-01 to 2010-03-31. The black dotted line shows the SLW fraction for the bulk cloud as observed by the CALIOP lidar, while the dashdotted line shows the same for cloud top, averaged over the period 2009-06-01 to 2013-05-31. The error bars correspond to one standard deviation in the lidar measurements.

cloud top ice, which is not as apparent in M92 due to the already excessive ice production. Another possible explanation could simply be that simulated ice crystals sediment at too high a rate from the cloud top to the interior of the cloud.

It should be stressed that while ice nucleation has been identified as an important area of improvement in climate models (Gettelman et al., 2023), there are other important cloud microphysical processes that are difficult to represent well, and improvements in cloud phase representation through INP parameterizations may be compensated by other model deficiencies. Ice crystal sedimentation (Gettelman et al., 2023) is one example, but the lack of relevant secondary ice production mechanisms is another, with recent work by Sotiropoulou et al. (2024) showing that no INP scheme can create a realistic cloud microphysical structure without secondary ice production. Implementing consistent secondary ice production schemes remains, however, an ongoing challenge (Sotiropoulou et al., 2024). For now, we can state that A21 improves the SLW fractions in Arctic clouds, performing best for bulk cloud and coming closer to observations at cloud top, despite the overestimation in cloud top SLW fraction at cold temperatures below $\sim$ -25°C.

### 3.2.3 Comparison to observed cloud cover and its phase partitioning

As ice number concentrations decrease, and cloud glaciating processes are reduced, we expect to see a longer cloud lifetime and increased cloud fraction. Overall, there is an annually averaged 0.016 increase in cloud fraction in the Arctic, which is largest in boreal autumn and winter (see Fig.B1c). This average increase manifests in the vertical as a cloud fraction increase

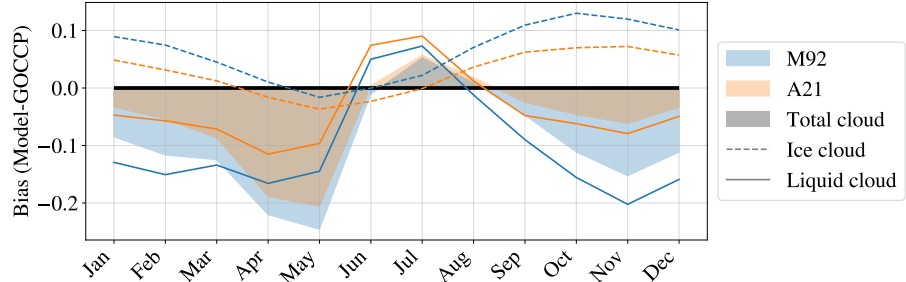

**Figure 8.** Cloud fraction biases between the model simulations A21 (orange) and M92 (blue), and CALIPSO-GOCCP observations, in absolute values. The figure shows the biases for total cloud cover (shaded), ice cloud fraction (dashed line) and liquid cloud fraction (solid line). The values are averaged over the area between 66.5°N and 82°N, and over the period 2007-04-01 to 2010-03-31. Cloud fraction for undefined cloud phase is not shown.

below $\sim 760$hPa that is somewhat offset by a slight reduction in cloud fraction above (see Fig. B1a). The changes are not
uniformly distributed throughout the region. We see annual average increases in cloud fraction over Greenland and the sea ice covered areas, which we do not see over the Norwegian Sea or Baffin Bay (see Fig. B1c). Over these ocean areas, we instead see slight decreases in some months, which can also be seen in Fig. B1b. The decrease in cloud fraction is more difficult to interpret in these idealised conditions where sea surface temperatures are fixed. In future studies it would be interesting to study if this effect is still seen in simulations with interacting sea surface temperatures.

In Fig. 8, we compare these changes in cloud cover to the CALIPSO-GOCCP data product for the Arctic as a whole. As noted in Sect. 2.3, this data product is useful for model comparison, but does not represent the absolute truth. Both model experiments have too little simulated cloud cover compared to CALIPSO-GOCCP for most parts of the year, but this negative bias is substantially reduced with the cloud cover increase in A21. The bias reduction with A21 is greatest in autumn and winter, when M92 overestimates the ice cloud fraction but underestimates the liquid cloud fraction by between 0.1 and 0.2.
The negative cloud cover bias is largest in spring, and while it is reduced in A21 it still remains at around 0.2. In July, however, the simulations have a smaller positive cloud cover bias of around 0.05. This is due to an overestimation of liquid cloud in both simulations, and in A21 this positive bias increases slightly. It should be noted that the cloud fractions in Fig. 8 are from the satellite simulator COSP (Bodas-Salcedo et al., 2011), not the direct model output shown in Fig. 3a and Fig. B1.

### 3.3   Modelled cloud radiative effects and comparison with observed radiative fluxes

**3.3.1   Modelled cloud radiative effects**

The differences in shortwave cloud radiative effect at the surface between A21 and M92 across the Arctic, divided by season, can be seen in Fig. 9. The calculation of cloud radiative effect at the surface can be found in Sect. B2. As expected, the change is highly dependent on the solar zenith angle, giving large absolute values during the sun-rich summer and late spring months, and close to zero changes for the winter months and at latitudes above 80°N in autumn. In summer, the change ranges from 2

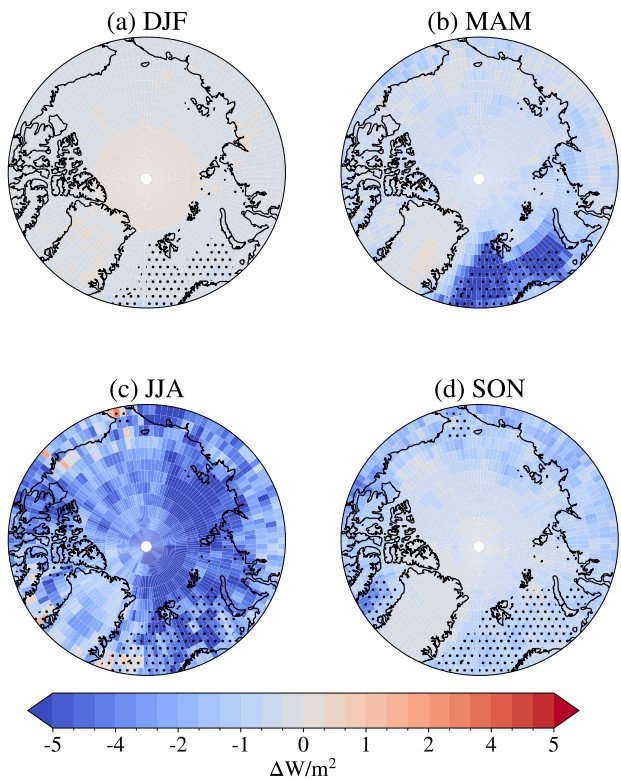

**Figure 9.** Differences in shortwave cloud radiative effect at the surface between A21 and M92 by season (averaged over the period 2007-04-01 to 2010-03-31). Negative values correspond to more solar radiation being reflected by the clouds in A21 compared to M92. Areas with open ocean area $\geq 85\%$ for more than 50 % of the year are hatched with black dots.

to 8 Wm$^{-2}$ less incoming solar radiation in A21 over the entire Arctic, consistent with the pattern of CLWP change in summer seen in Fig. 5. The average change across the Arctic is around -3 Wm$^{-2}$ in summer, which can be seen from Fig. 11a. In spring, the largest change is located over the Norwegian Sea, with values between -5 and -8 Wm$^{-2}$. This is also the region where we find the largest increase in spring CLWP. In autumn, the change is around -1 to -2 Wm$^{-2}$ at latitudes below 80° and Greenland, where the change is close to zero. This corresponds well with the distribution of CLWP change as well, taking

into account that there is much less solar radiation in autumn than in spring and summer and a significantly lower increase in CLWP over Greenland in autumn compared to the rest of the Arctic in A21. These changes culminate in an average relative change in surface shortwave cloud radiative effect across the Arctic of around -15 % (Fig. 11a).

The seasonal change in surface longwave cloud radiative effect between A21 and M92 across the Arctic is seen in Fig. 10. The increase in the longwave cloud radiative effect due to increased CLWP is non-linear. The longwave cloud radiative

effect is dependent on the cloud's longwave emissivity, which can be highly sensitive to changes in CLWP if it increases from previously small values, but is insensitive if the previous value was large. This is in contrast to the shortwave cloud radiative

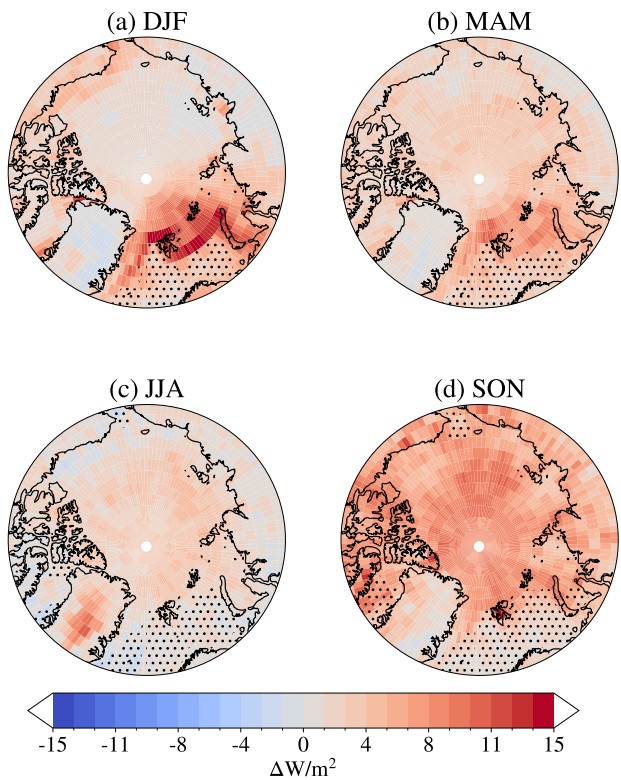

**Figure 10.** Differences in longwave cloud radiative effect at the surface between A21 and M92 by season (averaged by season over the period 2007-04-01 to 2010-03-31). Positive values correspond to more downwelling longwave radiation from clouds to the surface in A21 compared to M92. Note that the colorbar extends to threefold the extent of the colorbar in Fig. 9. Areas with open ocean area $\geq 85\%$ for more than 50 % of the season are hatched with black dots.

effect, for which the relationship with CLWP is closer to linearity. Taking the longwave emissivity dependence into account, the changes in longwave cloud radiative effect at the surface can largely be explained by the changes in CLWP as well,in addition to the changes in cloud cover. Changes in the cloud vertical profile may also play a role. A rough estimation of the
cloud longwave emissivity change, divided by season, can be found in Fig. B2, with a description of the calculation in Sect. B3.

    In the winter months, the largest increases in longwave cloud radiative effect can be found along the ice edge west, north and east of Svalbard, with values approaching 10 to 15 $\mathrm{Wm^{-2}}$. The increase in this area is even more pronounced than the changes we might expect in cloud longwave emissivity (see Fig. B2), compared to the rest of the Arctic. This is likely due to the fact
that this area is cloudier than the rest of the Arctic, making it more affected by the emissivity changes. In spring, the increase in longwave cloud radiative effect is strongest north and east of Svalbard, which is also where we see the strongest changes in cloud longwave emissivity. It is interesting to note that we observe little change in the longwave cloud radiative effect over the Norwegian Sea, even though the change in shortwave cloud radiative effect is between -5 and -8 $\mathrm{Wm^{-2}}$ in spring. This can be

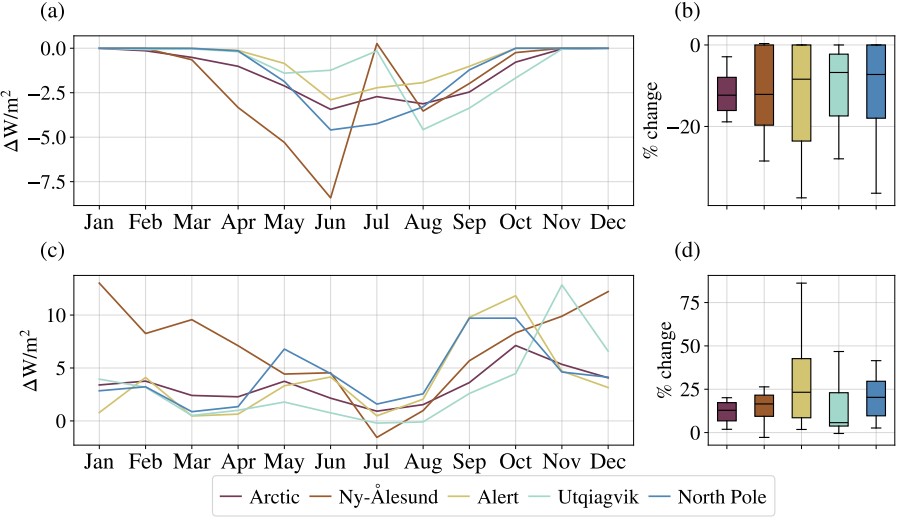

**Figure 11.** The change in total grid-box shortwave (a-b) and longwave (c-d) cloud radiative effect at the surface between A21 and M92, averaged over the period 2007-04-01 to 2010-03-31. Left: absolute change for each month averaged over selected regions. Right: the distribution of relative change over the months in the same regions.

explained by the clouds in this area already containing a fair amount of liquid water, so the resulting change in CLWP makes little difference to the cloud longwave emissivity (Fig. B2), while still having an effect on the cloud's reflection of shortwave radiation. In summer, what stands out is the comparatively large increase of longwave cloud radiative effect, between 5 and 10 $Wm^{-2}$, over Greenland. This corresponds to a large increase in cloud longwave emissivity, which is absent for the rest of the Arctic. The small increases in longwave cloud radiative effect we see over the sea ice covered Arctic might result from the general increase in low-level clouds (Fig. B1). In autumn, the increase is large (between 5 and 10 $Wm^{-2}$) over the entire Arctic, except for over the Norwegian Sea and parts of Greenland, where the CLWP increase is low (Fig. 5).

The total relative change in longwave cloud radiative effect across the Arctic is around 15 %, as seen in Fig. 11d. In Ny-Ålesund, the changes are largest in winter, while Alert and the North Pole, as well as the Arctic in general, show the largest changes in autumn, as well as a minor peak in spring. While the magnitude of change is comparable in Ny-Ålesund, approaching 15 $Wm^{-2}$ increased longwave cloud radiative effect in December, the average relative changes are largest over Alert and the North Pole.

### 3.3.2 Comparison with observed radiative fluxes

As the changes we see in surface longwave cloud radiative effect are quite large, it is natural to wonder whether the new state in A21 actually corresponds to a plausible climate. The cloud effect on outgoing longwave radiation at the TOA is negative, and most so during late summer and autumn. In Fig. 12a, we see the estimated cloud effect on outgoing longwave TOA flux from satellite measurements. First and foremost, we see that both M92 and A21 fall within the uncertainty estimates of CERES,

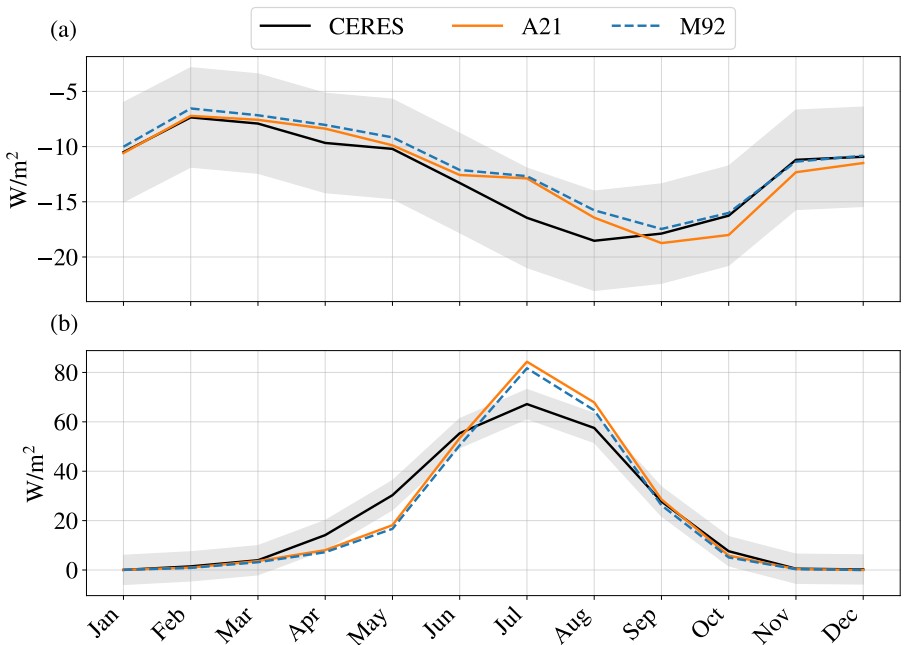

**Figure 12.** Cloud radiative effect on outgoing longwave (a) and shortwave (b) radiation at the top of the atmosphere, as estimated by CERES (black line) and modelled by NorESM2 in M92 (blue line) and A21 (orange line), averaged over the period 2007-04-01 to 2010-03-31. Gray shaded area corresponds to the CERES uncertainty.

indicating that as far as available measurements are concerned, the model representations fall within what we consider realistic values of radiation fluxes. While the difference between M92 and A21 at TOA is smaller than at the surface, A21 predicts a more negative cloud effect on outgoing longwave flux, as expected. This is especially clear in September and October each year, when the largest change in CLWP takes place. While the largest change in cloud longwave emissivity is estimated to be in winter and autumn (see Fig. B2), the autumn likely shows larger longwave cloud radiative effect changes due to the combination of high emissivity change and a larger amount of outgoing longwave radiation to intersect.

In Fig. 12b, we see the cloud effect on outgoing shortwave radiation at the TOA, which is a larger positive number the more clouds reflect solar radiation back to space. Here we see that there is a positive bias in summer and a smaller negative bias in spring. While the negative bias in spring is slightly improved in A21, the positive bias in summer is slightly enhanced. This is most likely connected to the summer cloud cover bias in the model, as also found by Shaw et al. (2022). Since liquid clouds are already overestimated in summer in M92, with its corresponding high primary ice production, reducing the primary ice production in A21 is not able to improve the cloud cover bias nor the bias in shortwave cloud radiative effect. While we do expect Arctic INP concentrations to be higher in the summer, as previously discussed in Sect. 3.1, is seems likely that this is a model issue that requires further improvement of other cloud processes.

In Fig. 13, we consider the radiation fluxes at the surface level by revisiting the Arctic areas we have examined earlier. The measurements are point-observations, without a possibility to isolate cloud radiative effects, and the corresponding model grid-boxes are quite large and cover varied surfaces. We therefore investigate the net radiation flux at the surface in order to compare the model experiments and radiation measurements more easily. Our focus is, however, first and foremost on longwave fluxes in winter and autumn, when the shortwave fluxes play less of a role. For all the stations, we see that the magnitude of the

negative net surface flux in winter is overestimated by the model simulations, pointing to a negative net radiation bias in the model in winter. However, this bias is smaller in A21 than M92. The biggest differences between the model experiments occur in Ny-Ålesund, where changes in cloud radiative effect clearly bring A21 more in line with observed winter-time flux. From this, we can draw the conclusion that the cloud radiative effect simulated in A21 brings NorESM2 closer to realistic radiation flux values, albeit without compensating completely for radiation biases. It is interesting to note that the new parameterization

reduces the gap between observed and modelled radiative fluxes the most in Ny-Ålesund, where we also expect our INP measurements to be most representative due to their similarity with Li et al. (2022).

To give an indication of the implications of these changes in cloud radiative effects, the changes in surface temperature between M92 and A21 across the Arctic domain are included in Fig. 14c, together with the net cloud radiative effect change throughout the year in Fig. 14a. It should be stressed that the surface temperature changes shown here are not representative

of the full response to the radiation changes, as the sea surface temperatures are fixed in the model, and also, the temperature changes over land and sea ice will be muted due to the fixed sea surface temperatures. For the Arctic in general (the blue line in Fig. 14c), we can see an average annual increase in A21 compared to M92 of $0.3°$C, with a maximum in October of $0.7°$C. This corresponds to an October net surface flux increase of 6 $\mathrm{Wm}^{-2}$. The largest temperature changes are found over the North Pole and Alert, of between $1.5°$C and $2°$C. Ny-Ålesund and Utqiagvik show similar increases as the former areas in

the magnitude of net surface flux, but due to the closer proximity to open ocean with fixed temperature, these areas likely show a more muted response in these simulations. This likely also explains why the North Pole shows a slightly larger temperature response than Alert. To determine the full temperature response of such a parameterization change, model experiments with dynamic oceans are necessary. These results, however, show that using observationally constrained INP concentrations has a large impact on the simulation of Arctic surface climate, and that using more realistic concentrations can be an important step

in improving model simulations.

## 4   Conclusions

In this study, we have found that Arctic clouds and their radiative effect are indeed sensitive to the INP parameterization, as shown in previous work by e.g. Xie et al. (2013). In attempting to constrain INP concentrations in the NorESM2 model using Arctic observations, we have quantified INPs active in the immersion freezing mode in Andenes in March 2021. Contrary to

DeMott et al. (2010), but in line with previous Arctic measurements (Li et al., 2022; Sze et al., 2023), we observed negligible correlations between INP freezing temperatures and the presence of ambient aerosols with diameter $\geq$ 0.5 μm. Indeed, temperature was the best indicator of INP concentrations that we could find, underscoring perhaps that the Arctic INP pop-

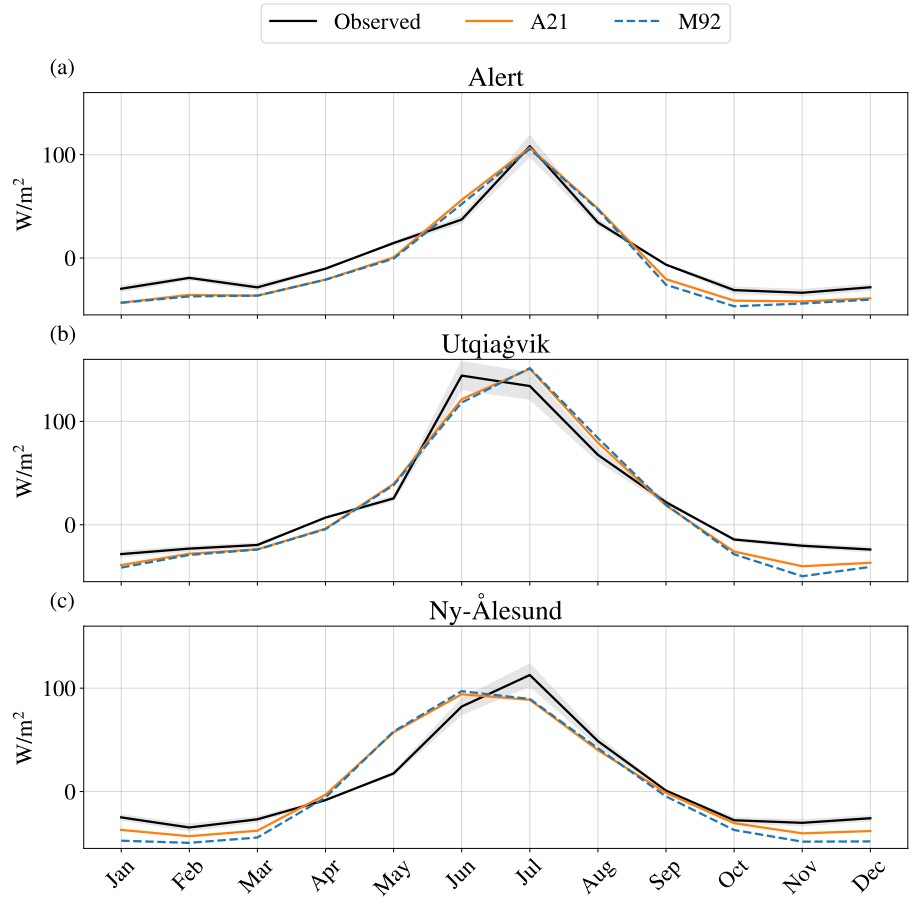

**Figure 13.** Net surface flux at various stations, from BSRN point-observations on the ground (black line) and corresponding grid-box values modelled by NorESM2 in M92 (blue line) and A21 (orange line), averaged over the period 2007-04-01 to 2010-03-31. Gray shaded area corresponds to the instrument uncertainty standards.

ulations differ in composition from those at lower latitudes in a way that makes them more difficult to capture with general INP schemes. The parameterization of the concentration as a function of temperature (A21) deviates two to four orders of magnitude from the Meyers et al. (1992) (M92) parameterization, designed for lower latitudes, but shows close similarity to the measurements of Li et al. (2022), taken at Ny-Ålesund in Fall 2019 and Spring 2020. From comparison with measurements in northern Greenland, however, we see that our measured concentrations are higher than the Greenlandic winter average, but slightly lower than the summer average (Sze et al., 2023). This could be an indication that our measured concentrations are on the higher end of what we would expect in winter in the rest of the Arctic, further from open ocean, but this remains to be investigated. While our measurements are confined to the ground, recent airborne measurements by Raif et al. (2024) have shown the presence of higher ice-nucleating particle concentrations at cloud level, underscoring the importance of investigat-

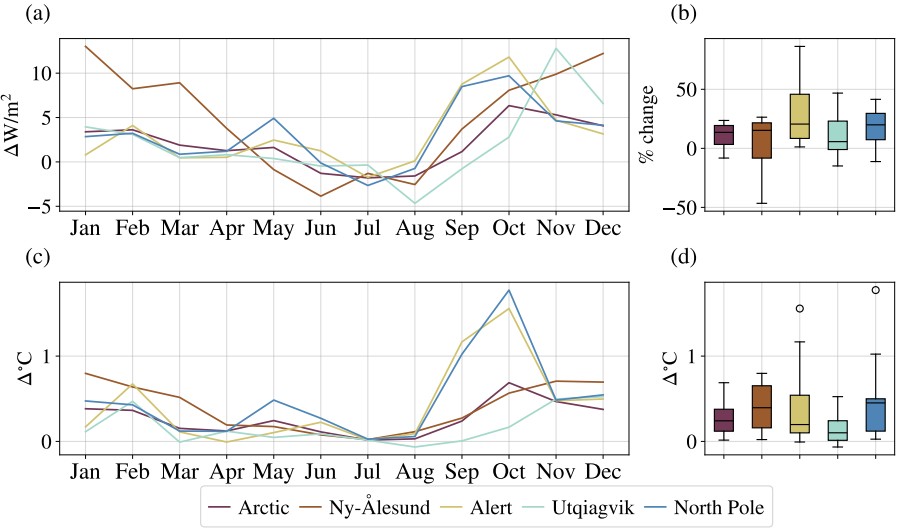

**Figure 14.** The change in net cloud radiative effect (a-b) and surface air temperature (c-d) between A21 and M92, averaged over the period 2007-04-01 to 2010-03-31. Left: absolute change for each month averaged over selected regions. Right: the distribution of relative change over the months in the same regions.

ing the vertical distribution of INPs, something which has also been found in other study areas by e.g. Knopf et al. (2023) and Moore et al. (2024).

The implementation of an observation-based INP parameterization for the Arctic in NorESM2 was shown to make a sub-465 stantial impact on the ice and liquid content of low-level Arctic clouds. The total grid-box CIWP was reduced by 14 % over the entire Arctic, the absolute changes being largest in boreal autumn, followed by winter. The total grid-box CLWP correspondingly increased by 70 % over the entire Arctic, with absolute changes in autumn of around 20 $gm^{-2}$. This led to a significant improvement in modelled SLW fractions for each cloud isotherm compared to spaceborne lidar observations using CALIOP, and an overall reduction in cloud cover bias for most of the year.

The same implementation led to increases in longwave cloud radiative effect at the surface in autumn, winter, and spring due to increased cloud optical thickness that dominated the decrease in shortwave cloud radiative effect at the surface in summer. The result led to an average increase in net cloud radiative effect at the surface of 2 $Wm^{-2}$ over the entire Arctic, with more than a 5 $Wm^{-2}$ increase in October and November. While these changes are quite large, they generally improve the simulated radiative fluxes compared to observations, apart from a bias increase in summer shortwave cloud radiative effect at TOA.

This bias increase suggests that the new INP parameterization in A21 may be too low in summer, although the cloud cover bias most likely requires improvement of other cloud processes as well. The changes in the cloud radiative effect showed an overall increase in average annual surface temperature of 0.3°C, with a maximum of 0.7°C in October, but which can only be considered a lower limit of a possible temperature effect as the sea surface temperatures are fixed in the model simulation.

The large increase in surface longwave radiative effect compared to shortwave is likely due to the non-linear relationship between cloud longwave emissivity and CLWP, causing the longwave emissivity to increase substantially as the CLWP increased from low values in autumn, winter and spring. The same non-linear relationship could matter for the role of cloud phase in future Arctic climate. With warming, we expect to see CLWP increases in Arctic clouds, and perhaps increased cloud radiative effect at the surface as the clouds become optically thicker. Exactly how warming will affect the cloud radiative effects, however, is not given. If we imagine a warming Arctic starting from a cloud ice content similar to M92, the non-linearity of longwave emissivity could perhaps create a larger increase in longwave cloud radiative effect than if we start from conditions similar to A21, as found by Tan and Storelvmo (2019). How such effects will be, however, is the outcome of a delicate interplay between cloud microphysical processes and climate change, about which there are still multiple uncertainties.

While this study supports that more realistic and regionally tailored INP parameterizations in climate models could be an important step for improved climate simulations, it also makes evident the pressing question of what role INPs will play in the future. Answering this question requires INP parameterizations that are not only latitude-dependent functions of temperature, such as ours, but the ones that are responsive to changes in relevant environmental factors based on physically established relationships. They need to be able to represent seasonal and spatial variations as well as warming-induced changes to the environment. Predicting the future effect of INPs also depends on being able to represent other relevant cloud microphysical processes, such as secondary ice production, in a satisfactory way. These will be important steps towards a better understanding of the role cold clouds play in Arctic and global warming and limiting the uncertainty in climate predictions (Prenni et al., 2007; Murray et al., 2021).

*Code and data availability.* The data sets produced for this study, both INP measurements and modelling results, can be found at https://doi.org/10.5281/zenodo.11617774, along with analysis scripts and model setup for the Norwegian Earth System model. The CALIOP L2 data used to derive SLF metrics and the CERES EBAF data can be downloaded freely at https://search.earthdata.nasa.gov/. The CALIPSO-GOCCP data product can be downloaded from https://climserv.ipsl.polytechnique.fr/cfmip-obs/Calipso_goccp.html. The surface radiation flux can be downloaded freely at https://www.pangaea.de/. The ERA5 data used to produce the back trajectories can be found at https://doi.org/10.24381/cds.bd0915c6. The surface air pressure observations in Andenes can be downloaded from https://thredds.met.no/thredds/catalog/met.no/observations/surface/87110/178/catalog.html. The colormap from Crameri et al. (2020) was used when preparing the figures.

## Appendix A: Ice-nucleating particle observations

### A1 Aerosol measurements

The relationships between INP freezing temperatures and ambient aerosols with diameter $\geq 0.5$ µm is shown in Fig. A1.

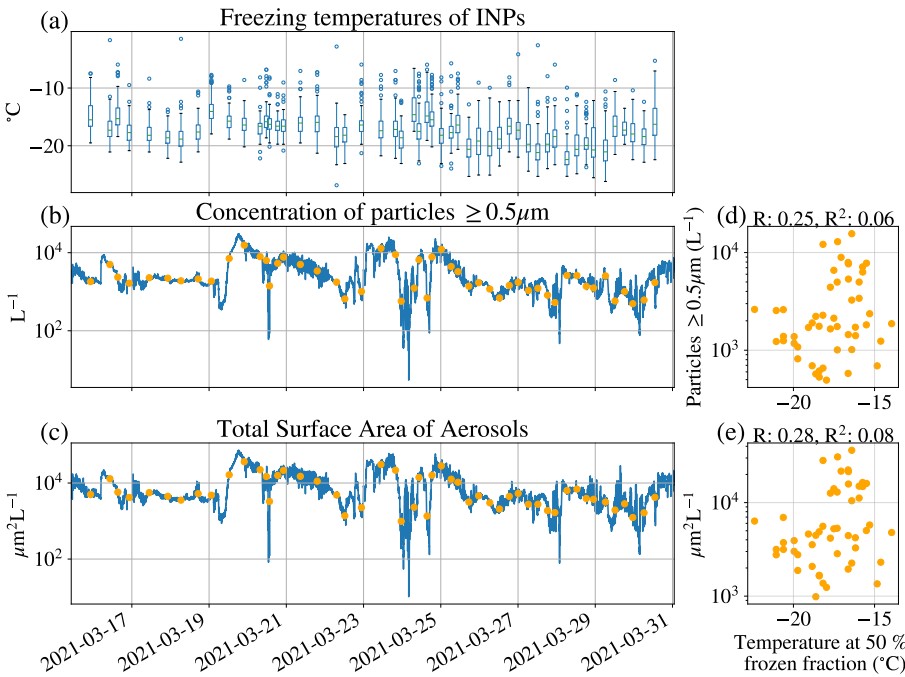

**Figure A1.** Freezing temperatures of all INP measurements in Andenes 2021 (a), compared with simultaneous measurements of ambient concentrations of aerosols with diameter ≥ 0.5 μm (b) and with total surface area of aerosols with diameter ≥ 0.5 μm (c). The orange dots in (b) show the average values over the INP sampling period, which are used for relationship estimation between the temperature at 50 % frozen fraction with larger aerosols in (d) and total aerosol surface area of these aerosols in (e).

## A2   Air parcel back trajectories

The potential source regions of INPs were identified from an analysis based on kinematic backward trajectories computed with LAGRANTO (Sprenger and Wernli, 2015). Using 3-hourly ERA5 reanalysis fields (Hersbach et al., 2023), air parcels were traced backwards from the location of Andenes for 10 days. Potential INP source locations where then extracted from the backward trajectories by finding locations where air masses arriving at low levels over Andenes were in potential contact with the surface during the last 3 days before arrival. To this end, trajectory points were identified by (i) selecting air parcel trajectories that were arriving with a pressure difference of less than 20 hPa to the surface pressure (corresponding to a height of ∼500 m above ground) at Andenes at the time of arrival, and (ii) along these trajectories, extracting the locations during the last 72 h before arrival where the air parcel was within 20 hPa of the surface pressure. Trajectories were only considered for those 3 h time windows where an INP sample had been taken. A map of these locations shows a concentration of potential INP source locations in the vicinity of the measurement site. Most locations are within the Norwegian Sea, with some branches going towards Greenland Sea and Fram strait, the Iceland Sea, and towards the North Atlantic (Fig. A2).

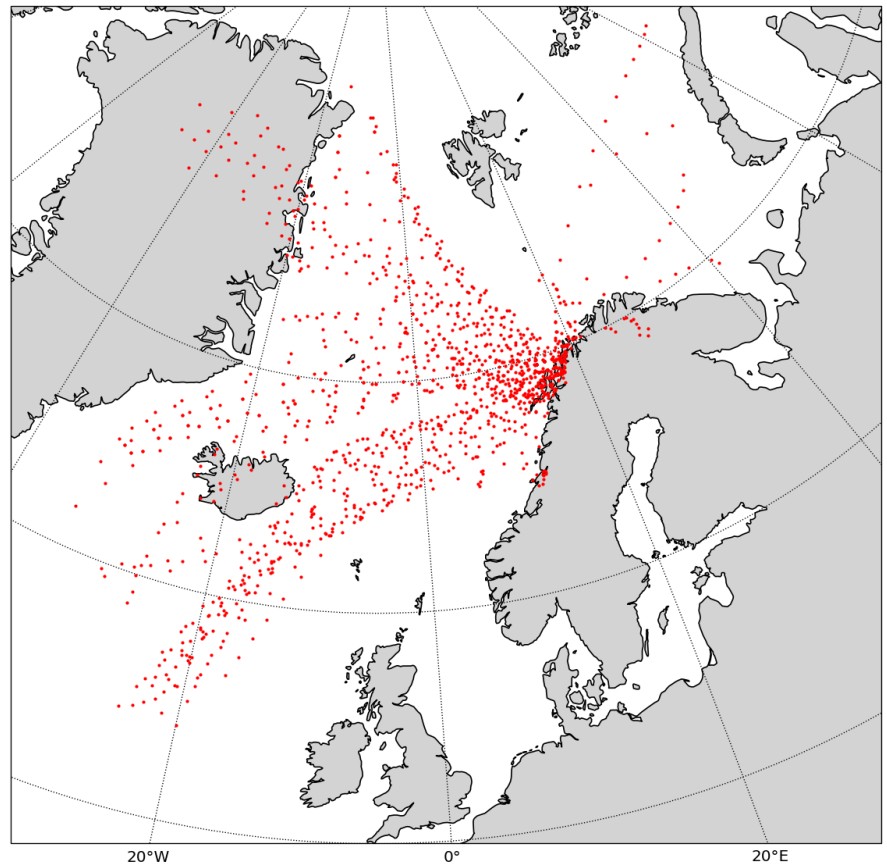

**Figure A2.** Air parcel back trajectory locations for 72 hours prior to INP measurements times, shown for when the air parcel was within 20 hPa of the surface pressure, or approximately 500 m height above ground. The trajectories are restricted to those arriving within 20 hPa from the surface pressure.

## Appendix B: Modelled cloud and radiation properties

**B1    Cloud cover**

The modelled cloud fraction changes are shown in Fig. B1.

**B2    Calculation of cloud radiative effect**

The modelled cloud radiative effect (CRE) at the surface for shortwave (SW) and longwave (LW) radiation is calculated as

$$\mathrm{SW_{CRE,surface}} = \mathrm{SW_{net,surface}} - \mathrm{SW_{net,clearsky,surface}} \qquad (B1)$$

$$\mathrm{LW_{CRE,surface}} = \mathrm{LW_{net,surface}} - \mathrm{LW_{net,clearsky,surface}}, \qquad (B2)$$

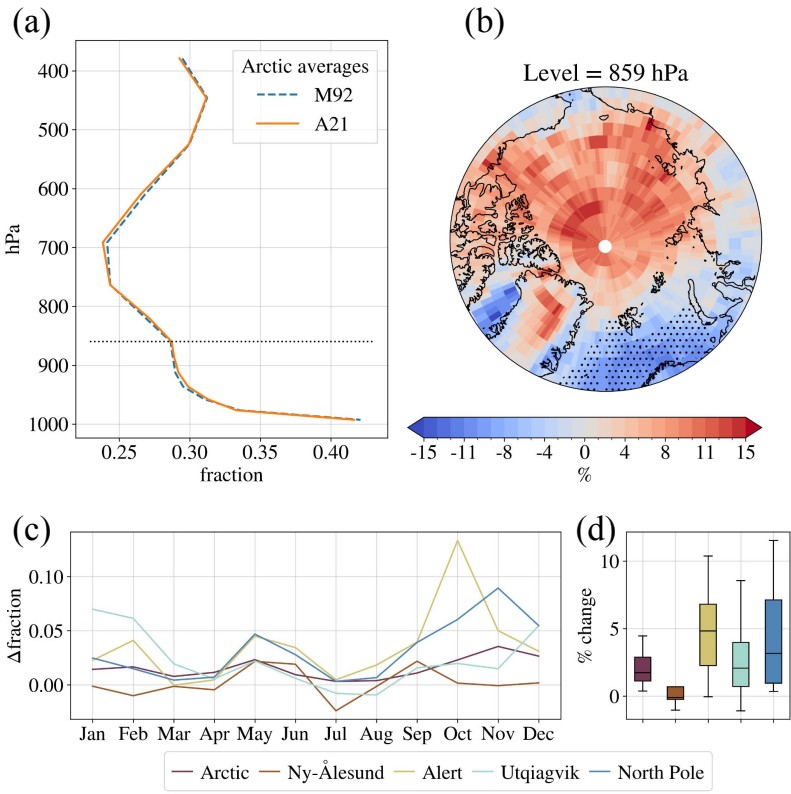

**Figure B1.** The cloud fraction for the M92 and A21 model experiments, averaged over the period 2007-04-01 to 2010-03-31. (a): the average values over all latitudes above 66.5°N for height levels in hybrid sigma pressure coordinates (midpoint). (b): the relative change from M92 to A21 at pressure level 859 hPa. The level is marked with a black dotted line in the left plot. (c) shows the total vertically-integrated cloud fraction. Left: absolute change for each month averaged over selected regions. (d): the distribution of relative change over the months in the same regions.

where downwelling radiation is defined as positive for both quantities. The modelled cloud radiative effect (CRE) at TOA for longwave (LW) radiation is calculated as

$$LW_{CRE,TOA} = LW_{out,TOA} - LW_{out,clearsky,TOA} \qquad (B3)$$

where upwelling radiation at TOA is defined as positive.

**B3 Cloud longwave emissivity**

To get a rough estimate of the downward cloud longwave emissivity (CLWE) we can expect from the simulated cloud liquid water path (CLWP), we calculate the CLWE based on the total grid-box CLWP as

$$CLWE = 1 - \exp(-k \times CLWP), \qquad (B4)$$

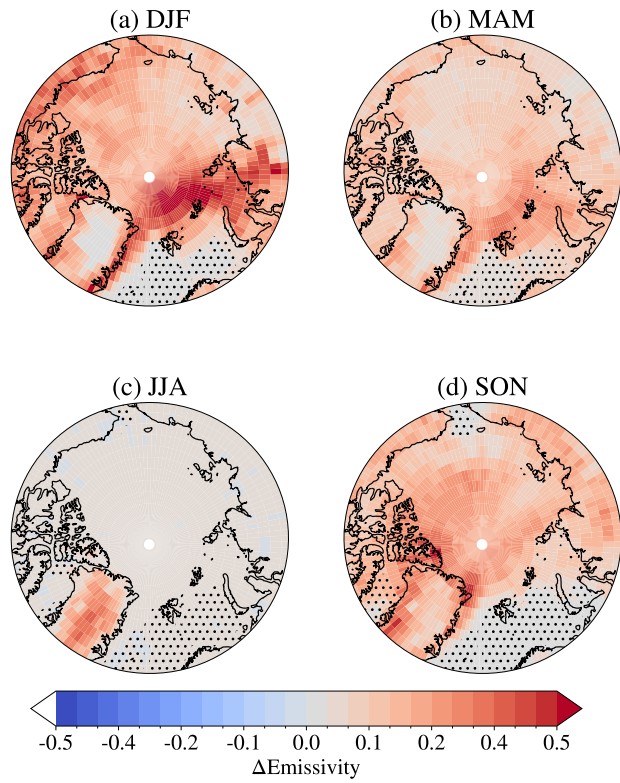

**Figure B2.** Differences in estimated cloud longwave emissivity between A21 and M92 by season (averaged by season over the period 2007-04-01 to 2010-03-31). Positive values correspond to higher longwave emissivity of clouds in A21 compared to M92. Areas with open ocean area $\geq 85\,\%$ for more than $50\,\%$ of the season are hatched with black dots.

where $k$ is the mass absorption coefficient. We use $k = 0.158$ m$^2$g$^{-1}$, the standard US value found by Stephens (1978). It
should be noted, however, that this value may vary slightly from place to place. The estimated CLWE changes can be found in
Fig. B2.

### B4 Total grid-box cloud water path

Below we present the total cloud water path in the model simulation with updated Arctic INPs (i.e. in experiment "A21"). The
total cloud water path is found by summing the total grid-box cloud ice water path and the total grid-box cloud liquid water
path.

### B5 Total precipitation

In Fig. B4 we present the difference in total precipitation between M92 and A21. This is the sum of large-scale and convective
precipitation, and is presented with the unit "mm month$^{-1}$", meaning mm per 30 days.

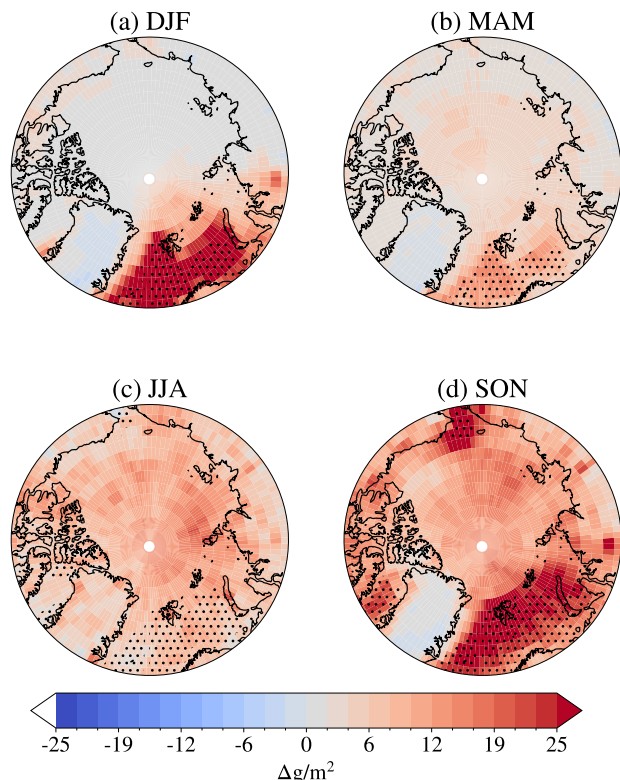

**Figure B3.** Differences in total grid-box cloud water path between A21 and M92 by season (averaged by season over the period 2007-04-01 to 2010-03-31). Areas with open ocean area $\geq 85\%$ for more than 50 % of the season are hatched with black dots.

*Author contributions.* Authors ABG, ROD, TC and FH all participated in performing INP measurements. Authors ROD, TC, FH and TS planned and implemented the measurement campaign, and outlined the measurement strategy, together with HS. ABG ran all model simulations, with assistance and contributions from TS, SH and ZM. SH and ZM provided model code modification. ABG, SH, TC and HS produced the figures. ABG wrote the manuscript, together with TS and ROD. All authors reviewed and commented on the manuscript.

*Competing interests.* We declare that we have no competing interests.

*Acknowledgements.* This work was supported by the European Research Council (ERC) through Grant StG 758005, as well as Grant CoG 101045273. ROD would also like to acknowledge support from EEARO-NO-2019-0423/IceSafari, contract no. 31/2020 under NO Grants 2014–2021 (EEA Grants/ Norway Grants) and the EU-HORIZON-WIDERA-2021 Grant no. 101079385. HS acknowledges support from the European Commission within the Horizon 2020 programme (Grant no. 773245). Resources for simulations and data storage were provided by UNINETT Sigma2, the National Infrastructure for High Performance Computing and Data Storage in Norway. We acknowledge all those

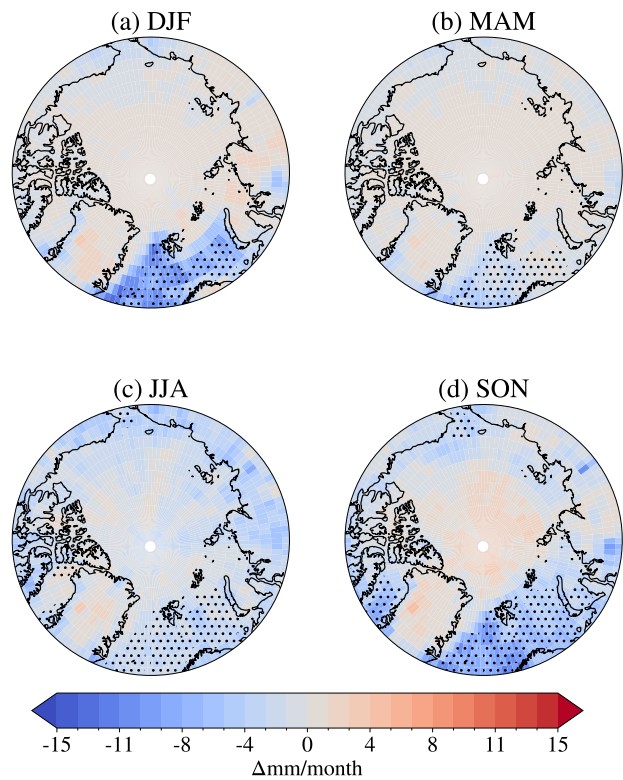

**Figure B4.** Differences in total precipitation between A21 and M92 by season (averaged by season over the period 2007-04-01 to 2010-03-31). Areas with open ocean area $\geq 85\%$ for more than 50 % of the season are hatched with black dots.

involved in the joint measurement campaign between MC2 and ISLAS. This includes Alena Dekhtyareva, Aina Marie Johannessen, Britta
Schäfer, Andrew Seidl and Iris Thurnherr, in addition to the aforementioned authors. Iris Thurnherr is especially acknowledged for her role
in calculating air parcel trajectories during the field campaign. The campaign was facilitated by Andøya Space. We would also like to thank
Dr. Jörg Wieder and Dr. Michael Rösch from ETH-Zürich for providing key equipment for the aerosol measurement setup. Finally, we would
like to thank the University of Oslo, Department of Geosciences Cold Climate Container (C3) infrastructure for providing the lab facilities
for DRINCO. Thanks to the two anonymous reviewers for useful comments that helped improve the manuscript.

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
