# Peer review of "Using a region-specific ice-nucleating particle parameterization improves the representation of Arctic clouds in a global climate model"

_EGUsphere, 2024_

## Referee Comment (RC3)

Comments on "Using a region-specific ice-nucleating particle parameterization improves the representation of Arctic clouds in a global climate model" by Gjelsvik et al.

General comments:

The authors use a recent temperature dependent INP parameterization constrained by the observations from a field campaigns in Norway. They found that the recent parameterization with new constrains causes a CLWP increases, which is better consistent with observations. The cloud induced surface fluxes changes are also larger than before and better agree with observations. Overall, I think this is very nice work. The overestimation of INP and subsequently induced uncertainties is necessary to be examined and documented. I recommend publishing it after addressing the comments below.

Major:

- The authors show that using the observationally based A21 parameterization induce CLWP increases and associated with changes of surface flux and air temperature, compared with Meyers et al., 1992. The constrains comes from the measurements in Andenes, March 2021. How do you justify that measurements at one cite are representative to the entire Arctic region? Here is another related question. The observations in Figure 2 align with the measurements form Li et al., 2023 and Sze et al., 2023 very well. Do you think similar overall and spatial changes could be induced if deriving the parameterization using their measurements? If so, do you think it better to constrain the parameterizations using as many as observations occurring in Arctic as the next step? If not, what do you think the potential reason that other measurements in high latitudes show different results?
- The authors mentioned in many places that the surface temperature is fixed. However, the interplay with surface temperature is very important in examining the cloud effects. Why not use the option with interaction surface temperature?

Minor:

- Line 146-147: why do you turn off the ice detrainment? How does that influence the results?
- Line 153-154: how do you justify that the deposition and condensation freezing are negligible for this specific case?
- Line 229-230: Why does the total aerosol surface area and INP freezing temperatures show a low correlation? The INP concentration depends on the temperature based on Figure 2.
- Line 235-236: Clarify how you implement the parameterization with temperature dependence only.
- Line 246-247: how do you infer that Figure 3a implies two different cold cloud populations? What are the two populations?
- Line 266-269: Did you evaluate the decreases in CIWP with observations? Does this seasonally variations leads to better agreement with observations?
- Some paragraphs have indents in the first line but others not. Better to be consistent.

---

## Author Response (AR1)

**Response to anonymous referee # 1**

*Dear referee,*

*Thank you very much for the valuable response to the article. We appreciate your comments, and have done our best to address them in the revised version of the manuscript. Below, we list your questions and comments, giving a response to them individually (in bold italics) and presenting the relevant changes to the manuscript on the line numbers in the revised version (marked in green).*

General comments:
The authors use a recent temperature dependent INP parameterization constrained by the observations from a field campaigns in Norway. They found that the recent parameterization with new constrains causes a CLWP increases, which is better consistent with observations. The cloud induced surface fluxes changes are also larger than before and better agree with observations. Overall, I think this is very nice work. The overestimation of INP and subsequently induced uncertainties is necessary to be examined and documented. I recommend publishing it after addressing the comments below.

Major:
- The authors show that using the observationally based A21 parameterization induce CLWP increases and associated with changes of surface flux and air temperature, compared with Meyers et al., 1992. The constrains comes from the measurements in Andenes, March 2021. How do you justify that measurements at one cite are representative to the entire Arctic region? Here is another related question. The observations in Figure 2 align with the measurements form Li et al., 2023 and Sze et al., 2023 very well. Do you think similar overall and spatial changes could be induced if deriving the parameterization using their measurements? If so, do you think it better to constrain the parameterizations using as many as observations occurring in Arctic as the next step? If not, what do you think the potential reason that other measurements in high latitudes show different results?

*The representativeness of the measurements and the parameterization is an important issue we wish to address in more detail in the manuscript. We expect our measurements to be more representative of the Arctic than the measurements which form the base of the Meyers' parameterization, as these are from Wyoming and Manchester. However, there is a large variability in Arctic INPs from previous studies throughout the region in different seasons, which we do not capture with our limited measurement period at only one site.*

*The seasonal cycles in the Arctic INP, with higher INP concentrations in summer, are clear from measurements. This likely depends both on local sources like open ocean, local dust and terrestrial vegetation, as well as long-range transport. How much a site is influenced by various sources will create a substantial spatial variability.*

*Additionally, while our measurements and many of those we compare them to are at ground level, the picture might look different at cloud level. Recently published airborne measurements from Raif et al. (2024) show that airborne measurements can*

*exhibit quite high INP concentrations, which likely consist of biologically enriched dust.*

*To highlight these caveats with our measurements, we have extended lines 232-238 to the following paragraph:*

*Arctic glacial dust with high ice-nucleating ability has also been shown to play an important role in the lower troposphere during summer months (Tobo et al., 2019; Kawai et al., 2023, Tobo et al., 2024), in addition to the variability coming from low latitude dust sources throughout the year (Shi et al., 2022). It should be noted that our measurements are a snapshot in time. The seasonal Arctic INP cycle certainly contains both lower and higher INP concentrations compared to our measurements, both related to the aforementioned local sources as well as transport from lower latitudes. Importantly, our measurements are only at the surface level, while studies from Shi et al. (2022) and Raif et al. (2024) show that long-range transported dust can play an important role at higher altitudes.*

*We have also added the following on lines 459-462:*

*While our measurements are confined to the ground, recent airborne measurements by Raif et al. (2024) have shown the presence of higher INP concentrations at cloud level, underscoring the importance of investigating the vertical distribution of INPs, something which has also been found in other study areas by e.g. Knopf et al. (2023) and Moore et al. (2024).*

*The reason why we think Andenes is an especially interesting place to study INPs is because we are able to capture air masses both coming into and out of the Arctic. The parameterization based on our measurements may look slightly different if we combined more of the measurements we compare with, but as there are already strong similarities in the average INP concentration observations, we do not expect that incorporating more studies into our parametrization will yield significant differences, as long as we use one average for the entire Arctic. However, we think interesting future work would be to study the importance of the spatial and temporal heterogeneity of Arctic INPs, and their influence from sources proposed to be important Arctic INP sources based on previous measurements.*

- The authors mentioned in many places that the surface temperature is fixed. However, the interplay with surface temperature is very important in examining the cloud effects. Why not use the option with interaction surface temperature?

*In our model setup, the sea surface temperatures are prescribed, while the land and sea ice covered surface temperatures are able to respond to radiation changes. While these temperature changes are somewhat muted when the sea surface temperatures remain fixed,  we can learn a lot about cloud and radiation changes with these simulations. Critically, prescribed-SST simulations are the standard tool for calculating radiative forcings (Forster et al., 2021), including the cloud radiative effects shown in our Figs. 9-14. A fully interactive ocean would absorb the energy imbalance caused by the INP parameterization change, and hence prevent it from*

*cleanly being quantified. Interactive ocean models are more useful for fully capturing the surface temperature response to such a forcing, although this will bring into the results model uncertainties from uncertain climate feedbacks (i.e. change in surface temperature per unit radiative effect perturbation, which varies strongly across models). We hence use prescribed-SST simulations as a first step to understand how the simulated Arctic clouds are affected by observational INP constraints, as is the dominant practice in GCM studies of INP effects (e.g. Lohmann & Hoose, 2009; McGraw et al., 2020; Storelvmo et al., 2008). We do intend to test this parameterization with interactive ocean temperatures in future simulations, which will more comprehensively reveal INP impacts on Arctic climate, although this is beyond the goals of the present study. Further, such a fully coupled model run is substantially more expensive than the simulations shown herein, and requires longer simulations in order to distinguish the role of the parameterization change from the model noise.*

*To underline the importance of interacting sea surface temperatures for a full understanding, we have added the following sentence to the section on cloud cover changes (see lines 346-348):*

*The decrease in cloud fraction is more difficult to interpret in these idealised conditions where sea surface temperatures are fixed. In future studies it would be interesting to study if this effect is still seen in simulations with interacting sea surface temperatures.*

Minor:
- Line 146-147: why do you turn off the ice detrainment? How does that influence the results?

*In hindsight, this wording was inaccurate and did not precisely reflect the actual changes made in the model. To clarify, in CAM6-Nor, the phase of cloud particles being detrained from convective updrafts is governed by a linear temperature ramp, where all cloud particles below -35°C are ice, all particles above -5°C are liquid, and the fraction of detrained cloud ice particles decreases linearly as a function of temperature between these two temperatures. Following Hofer et al. (2024), we have adjusted this temperature ramp to better align the simulated supercooled liquid water (SLW) fractions in mixed-phase clouds with observations. In our case, all cloud particles with temperatures above -35°C are detrained as liquid, but can subsequently freeze as they are incorporated into the stratiform cloud microphysics. It should be stressed that this change is applied to both experiments, with and without the new Arctic INP parameterization.*

*We have changed lines 165-167, and moved it to Sect 2.2.3:*

*Additionally, the detrainment of cloud particles through convective updrafts is shifted from having the ice phase fraction of detrained particles decreasing linearly with temperature, to having all cloud particles detrained as liquid above -35°C, following Hofer et al. (2024).*

- Line 153-154: how do you justify that the deposition and condensation freezing are

negligible for this specific case?

*Several studies have found supercooled liquid water being a prerequisite for ice formation in mixed-phase clouds (see e.g. Ansmann et al., 2009, de Boer et al., 2011, Westbrook and Illingworth, 2011). Immersion freezing is therefore accepted as the dominant ice nucleation pathway in these clouds. It should be specified, however, that we only neglect deposition/condensation freezing for clouds in the temperature range -37°C to 0°C, so we still include deposition freezing in cirrus clouds. We have now tried to clarify this on lines 150-160:*

*In CAM5, the different heterogeneous ice nucleation pathways in mixed-phase clouds are parameterized independently, namely, contact freezing (Young, 1974), immersion freezing (Bigg, 1953) and deposition and condensation freezing (Meyers et al., 1992). Here, we update the Meyers et al. (1992) parameterization (hereafter: "M92") in the Arctic, using our measurements from Andenes. This parameterization is active in the temperature range -37°C to 0°C, and is responsible for more than 90 % of ice crystals formed in CAM5 mixed-phase clouds (English et al., 2014). Since the measured INP concentrations are relevant for the immersion mode, replacing the M92 parameterization with our measurements entails excluding deposition and condensation freezing in Arctic mixed-phase clouds. This exclusion is justified by observational studies that found deposition and condensation freezing to be negligible for mixed-phase clouds (Ansmann et al., 2009; Boer et al., 2011; Westbrook and Illingworth, 2011). However, our exclusion of deposition freezing does not apply to temperatures below -37°C (cirrus regime). As we update the M92 parameterization using our INP measurements in immersion freezing mode, we exclude the Bigg (1953) immersion freezing parameterization.*

- Line 229-230: Why does the total aerosol surface area and INP freezing temperatures show a low correlation? The INP concentration depends on the temperature based on Figure 2.

*Thanks for raising this important question which we hope that future studies will eventually answer. Our best guess is that we do not have a full understanding of the composition or aerosol species of the INPs we measure and as such, are unable to predict the size or surface area they should have. As these INPs represent a unique fraction of the general aerosol population, and this fraction is also likely source-region/air mass dependent, we do not expect the surface area of the general aerosol population to correlate with the rare aerosol that can act as INPs. This lack of a correlation between aerosol surface area and INP concentration has also been found in previous INP studies from the Arctic, e.g. by Li et al., 2022.*

*We have added the following sentence on lines 259-260 to address this further:*

*Explaining the variability of the INPs requires more knowledge of the particle composition than we have from our measurements.*

- Line 235-236: Clarify how you implement the parameterization with temperature dependence only.

*In light of this and the previous comment, we have now added the following update to the sentence on lines 263-265:*

*Due to the lack of correlation with aerosol ≥ 0.5µm, we implement the parameterization with temperature dependence only, using the exponential fit seen in Fig. 2 to represent the number of activated INPs in mixed-phase clouds as a function of temperature (see Sect. 2.2.3).*

*In Sect. 2.2.3. we write on lines 169-174:*

*The model experiments conducted in this study are referred to as "M92" and "A21". M92 is a CAM6-Nor set-up, following the ice production adjustments described above, but using the standard CAM5 heterogeneous ice nucleation schemes. In A21, the activated ice number produced by the parameterizations of Meyers et al. (1992) and Bigg (1953) is replaced if the latitude exceeds 66.5°N. For these latitudes, the immersion freezing INPs in the temperature range -37°C to 0°C is determined by the INP observations at Andenes instead. The new parameterization is a temperature-dependent exponential fit of the Andenes INP measurements (see Fig. 2).*

- Line 246-247: how do you infer that Figure 3a implies two different cold cloud populations? What are the two populations?

*Thanks for pointing this out and we agree that this was not well formulated in the text. What we wanted to express is that with the very large reduction in ice formation we expect that many of the clouds that were categorized as ice clouds in M92 are removed from the ice cloud category in A21. Looking at cloud ice number concentration in the small remaining ice cloud population in A21 thus gives us just a small part of the story of what has happened with cloud ice number concentrations. We have made a small update to attempt to clarify this.*

*Lines 276-278: Many of the ice clouds in M92 have been entirely transformed to liquid clouds in A21, which makes it less straightforward to compare in-ice-cloud quantities with M92.*

*To make it clear that we do not include supercooled liquid clouds in the cold cloud category, we have also changed from "cold cloud" to "ice cloud" in this section. We define it on lines 269-270 as:*

*Ice clouds are defined here as clouds with an ice mixing ratio larger than 10−6, and can also contain supercooled liquid water.*

- Line 266-269: Did you evaluate the decreases in CIWP with observations? Does this seasonally variations leads to better agreement with observations?

*We have not evaluated decreases in CIWP with observations, as most estimates of CIWP have a high uncertainty, particularly in the Arctic. However, in the revised*

version of this manuscript, we include an evaluation of the changes in bias in ice and liquid cloud fraction with respect to CALIPSO lidar measurements. With the new model parameterization, the ice cloud bias is reduced considerably from July to April due to less ice cloud, but it is increased slightly compared to M92 in May and June. One could say that the new parameterization leads to better agreement with observations in all seasons except summer, where there is already too much (liquid) cloud. However, the parameterization has a relatively minor impact in this season compared to the rest of the year, as expected.

We have changed Sect. 3.2 a bit, changing the title to Modelled cloud phase and comparison with observations and have added a section called Comparison to observed cloud cover and its phase partitioning at the end of the section.

There we include the description of the modelled cloud quantities, and add the following section on comparison to observations on lines 349-357, together with a new figure (Fig. 8) showing the comparison:

In Fig. 8, we compare these changes in cloud cover to the CALIPSO-GOCCP data product for the Arctic as a whole. As noted in Sect. 2.3, this data product is useful for model comparison, but does not represent the absolute truth. Both model experiments have too little simulated cloud cover compared to CALIPSO-GOCCP for most parts of the year, but this negative bias is substantially reduced with the cloud cover increase in A21. The bias reduction with A21 is greatest in autumn and winter, when M92 overestimates the ice cloud fraction, but underestimates the liquid cloud fraction by between 0.1 and 0.2. The negative cloud cover bias is largest in spring, and while it is reduced in A21 it still remains at around 0.2. In July, however, the simulations have a smaller positive cloud cover bias of around 0.05. This is due to an overestimation of liquid cloud in both simulations, and in A21 this positive bias increases slightly. It should be noted that the cloud fractions in Fig. 8 are from the satellite simulator COSP (Bodas-Salcedo et al., 2011), not the direct model output shown in Fig. 3a and Fig. B1.

We have also added the following updates to the conclusion on lines 466-468:

This led to a significant improvement in modelled SLW fractions for each cloud isotherm compared to spaceborne lidar observations using CALIOP, and an overall reduction in cloud cover bias for most of the year.

A description of the CALIPSO-GOCCP data product is added in the Methods section on lines 202-208:

Furthermore, we use observations of cloud fraction, ice cloud fraction and liquid fraction to evaluate the model simulations. In order to make a direct comparison, we use the GCM-Oriented Cloud-Aerosol Lidar and Infrared Pathfinder Satellite Observation (CALIPSO) Cloud Product (GOCCP, Chepfer et al., 2010), and compare it to the model output using the CFMIP Observation Simulator Package (COSP, Bodas-Salcedo et al., 2011). It should be noted that spatial averaging is required to

*make a direct comparison between satellite observations and the model, and that different cloud climatologies from remote sensing vary between themselves, making CALIPSO-GOCCP no absolute truth (Chepfer et al., 2010).*

*Lastly, we also present how the annually averaged total cloud water path corresponds with different reanalysis products, by adding the following on lines 309-312:*

*The annual mean total cloud water path north of 60°N is 53.0 gm−2 in A21 and 48.0 gm−2 in M92. Both of these values are on the lower end of the range in mean cloud water path (49.5 gm−2 to 82.7 gm−2) estimated by reanalysis products in this region (Gu et al., 2021). It should be noted that measurements of total cloud water path are known to have a large uncertainty at high latitudes (Khanal et al., 2020), particularly due to difficulties in separating precipitation and cloud particles.*

*We add a figure of the differences in total cloud water path between M92 and A21, by season, in the appendix, see Fig. B3.*

- Some paragraphs have indents in the first line but others not. Better to be consistent.

*Thank you for pointing this out. We have left the first paragraph after a title/section header without an indent following Chicago style (https://www.chicagomanualofstyle.org/qanda/data/faq/topics/ManuscriptPreparation/faq0223.html). Otherwise we have not been able to find any new paragraphs without indents, but please let us know if there is anything we have missed.*

***References:***

*Ansmann, A., M. Tesche, P. Seifert, D. Althausen, R. Engelmann, J. Fruntke, U. Wandinger, I. Mattis, and D. Müller (2009), Evolution of the ice phase in tropical altocumulus: SAMUM lidar observations over Cape Verde, J. Geophys. Res., 114, D17208, doi:10.1029/2008JD011659.*

*de Boer, G., H. Morrison, M. D. Shupe, and R. Hildner (2011), Evidence of liquid dependent ice nucleation in high-latitude stratiform clouds from surface remote sensors, Geophys. Res. Lett., 38, L01803, doi:10.1029/2010GL046016.*

*Forster, P., T. Storelvmo, K. Armour, W. Collins, J.-L. Dufresne, D. Frame, D.J. Lunt, T. Mauritsen, M.D. Palmer, M. Watanabe, M. Wild, and H. Zhang, 2021: The Earth's Energy Budget, Climate Feedbacks, and Climate Sensitivity. In Climate Change 2021: The Physical Science Basis. Contribution of Working Group I to the Sixth Assessment Report of the Intergovernmental Panel on Climate Change [Masson-Delmotte, V., P. Zhai, A. Pirani, S.L. Connors, C. Péan, S. Berger, N. Caud, Y. Chen, L. Goldfarb, M.I. Gomis, M. Huang, K. Leitzell, E. Lonnoy, J.B.R. Matthews, T.K. Maycock, T. Waterfield, O. Yelekçi, R. Yu, and B. Zhou (eds.)]. Cambridge University Press, Cambridge, United Kingdom and New York, NY, USA, pp. 923–1054, doi:10.1017/9781009157896.009.*

*Hofer, S., Hahn, L.C., Shaw, J.K. et al. Realistic representation of mixed-phase clouds increases projected climate warming. Commun Earth Environ 5, 390 (2024). https://doi.org/10.1038/s43247-024-01524-2*

*Li, G., Wieder, J., Pasquier, J. T., Henneberger, J., and Kanji, Z. A.: Predicting atmospheric background number concentration of ice-nucleating particles in the Arctic, Atmos. Chem. Phys., 22, 14441–14454, https://doi.org/10.5194/acp-22-14441-2022, 2022.*

*Lohmann, U., & Hoose, C. (2009). Sensitivity studies of different aerosol indirect effects in mixed-phase clouds. Atmospheric Chemistry and Physics, 9(22), 8917-8934.*

*McGraw, Z., Storelvmo, T., Samset, B. H., & Stjern, C. W. (2020). Global radiative impacts of black carbon acting as ice nucleating particles. Geophysical Research Letters, 47(20), e2020GL089056.*

*Raif, E. N., Barr, S. L., Tarn, M. D., McQuaid, J. B., Daily, M. I., Abel, S. J., Barrett, P. A., Bower, K. N., Field, P. R., Carslaw, K. S., and Murray, B. J.: High ice-nucleating particle concentrations associated with Arctic haze in springtime cold-air outbreaks, EGUsphere [preprint], https://doi.org/10.5194/egusphere-2024-1502, 2024.*

*Storelvmo, T., Kristjánsson, J. E., Lohmann, U., Iversen, T., Kirkevåg, A., & Seland, Ø. (2008). Modeling of the Wegener–Bergeron–Findeisen process—Implications for aerosol indirect effects. Environmental Research Letters, 3(4), 045001.*

*Westbrook, C. D., and A. J. Illingworth (2011), Evidence that ice forms primarily in supercooled liquid clouds at temperatures > −27°C, Geophys. Res. Lett., 38, L14808, doi:10.1029/2011GL048021.*

**Response to anonymous referee # 2**

*Dear referee,*

*Thank you very much for your valuable comments and important points that have helped us improve the quality of this manuscript. We have tried to answer your comments and questions below, by listing them, giving a response to them individually (in bold italics) and presenting the relevant changes to the manuscript on the line numbers in the revised version (marked in green).*

Review of "Using a region-specific ice-nucleating particle parameterization improves the representation of Arctic clouds in a global climate model" by Gjelsvik and co-authors.

In this study the authors use measurements of ambient ice nucleating particle (INP) concentrations from within the Arctic to assess the sensitivity of the NorESM Earth System Model to the representation of immersion mode heterogeneous ice nucleation. Ambient aerosols greater than 0.5 um in diameter were collected over a period of 15 days in March 2021 and processed in an instrument that measures the number of INP activated as a function of temperature and the volume of sampled air. Using these data an INP parameterization was derived for the representation of immersion-mode ice nucleation in supercooled cloud liquid droplets. The authors incorporated the parameterization into a cloud microphysics module in NorESM to replace primary ice production in the Arctic from the default mechanism(s). A comparison between simulations using the default (named M92) and the new setup (named A21) was used to assess whether the new setup improves the model's ability to reproduce observed quantities of the Arctic atmosphere. The new A21 setup improves the resulting bulk cloud phase partitioning between liquid and water, impacts the amount of cloud ice and cloud liquid water loadings, and reduces the bias apparent in radiative fluxes at the surface. The study is not entirely novel but does help extend our knowledge on the role of INPs in the climate, and specifically the Arctic region which is less understand than others. It also demonstrates, as have other studies, the sensitivity of simulated high-latitude clouds to the representation of ice production. However, I believe there are several major issues within the methodology and analysis that need addressing before the manuscript is in a position to be accepted for publication.

Major comments.

The main result of the study is that reducing the simulated INP concentrations results in more liquid water and less ice, which better matches remotely observed cloud-phase partitioning. One caveat to this result is that it could be compensating for other poorly represented processes in the cloud microphysics scheme. Are the authors confident that this is not the case? For instance, Gettelman et al. (2023) have demonstrated that Arctic LWP and IWP in CESM2 is sensitive to several different cloud microphysical processes. Do other models using the M92 parameterization (or similarly high primary ice production rates) suffer from the same issue? It may not be possible to answer these questions, but this caveat should be thoroughly addressed in the manuscript.

*We agree with the reviewer that there are several microphysical processes important for setting cloud properties, although we do hold that primary ice nucleation is a*

*sufficiently critical process to be our sole focus in this study on Arctic mixed-phase clouds. While the cited Gettelman et al. study (which two of our co-authors contributed to) indeed shows LWP and IWP to be sensitive to several model changes, those results are not inconsistent with our choice of focus here. Our main metric of observation-model comparison is not LWP or IWP, but rather cloud thermodynamic phase, which is specific to mixed-phase clouds. LWP and IWP we show to demonstrate the ramifications of the INP modification, rather than to verify model accuracy. LWP and IWP are columnar and hence depend on warm and cirrus clouds as well, so give a poorer indication if mixed-phase clouds are accurately represented. While most plots in Gettelman et al lump together cirrus and mixed-phase clouds, Fig. 7 is more revealing, as Arctic mixed-phase clouds can roughly be identified as low clouds >66.5 N. From this figure, it is apparent that these clouds undergo a consistent increase in ice phase fraction in all versions of the model with the ice nucleation change, and are less affected by any other assessed process (note: the changes at cirrus altitudes here likely reflect an issue in this diagnostic at high altitudes, as 7a shows upper tropospheric ice fraction is nowhere near the expected value of one). Arctic cloud phase is hence in those simulations largely a function of INPs. Further, in the Gettelman et al. study nearly all INPs in mixed-phase clouds are mineral dust. Because this species has a limited role in the Arctic, the influence of ice nucleation should likely be even more dominant in this region than it is shown in the Gettelman et al. study.*

*There absolutely are other poorly represented cloud processes in the model. This is an important point that we have attempted to stress further by adding the following paragraph on lines 330-338:*

*It should be stressed that while ice nucleation has been identified as an important area of improvement in climate models (Gettelman et al., 2023), there are other important cloud microphysical processes that are difficult to represent well, and improvements in cloud phase representation through INP parameterizations may be compensated by other model deficiencies. Ice crystal sedimentation (Gettelman et al., 2023) is one example, but the lack of relevant secondary ice production mechanisms is another, with recent work by Sotiropoulou et al. (2024) showing that no INP scheme can create a realistic cloud microphysical structure without secondary ice production. Implementing consistent secondary ice production schemes remains, however, an ongoing challenge (Sotiropoulou et al., 2024). For now, we can state that A21 improves the SLW fractions in Arctic clouds, performing best for bulk cloud and coming closer to observations at cloud top, despite the overestimation in cloud top SLW fraction at cold temperatures below ~ -25°C.*

"Improved representation of Arctic clouds". The authors have only selected a few observable quantities to assess the representation of Arctic clouds. There are others that could be considered to fully justify this statement – including LWP and IWP, cloud fraction, and precipitation. Widespread LWP increases of 30 gm-2 apparent in Figure 5 will surely be seen in the observations / model biases. Similarly, the cloud fraction changes in Figure B1 may be substantial enough to be seen in observations / model biases. Also, why was the TOA shortwave CRE not compared to observations (as for longwave in Figure 11)? I couldn't find anywhere in the text that explained their omission.

We agree and as such have extended the analysis with comparisons to cloud cover, ice cloud and liquid cloud fraction, and TOA shortwave CRE, also based on the comment from the first anonymous reviewer. We have not evaluated decreases in IWP or LWP with observations, as most observations of IWP/LWP have a high uncertainty. However, in the revised version of this manuscript we include an evaluation of the changes in bias in ice and liquid cloud fraction with respect to the CALIPSO-GOCCP data product, in addition to cloud cover in general. With the new model parameterization, the ice cloud bias is reduced from July to April due to less ice cloud, but it is increased slightly compared to M92 in May and June. One could say that the new parameterization leads to better agreement with observations in all seasons except summer, where there is already too much (liquid) cloud. However, the parameterization has a relatively minor impact in this season compared to the rest of the year, as expected.

We have changed Sect. 3.2 a bit, changing the title to Modelled cloud phase and comparison with observations and adding a section called Comparison to observed cloud cover and its phase partitioning at the end of the section.

There we include the description of the modelled cloud quantities, and add the following section on comparison to observations on lines 349-357, together with a new figure (Fig. 8) showing the comparison:

In Fig. 8, we compare these changes in cloud cover to the CALIPSO-GOCCP data product for the Arctic as a whole. As noted in Sect. 2.3, this data product is useful for model comparison, but does not represent the absolute truth. Both model experiments have too little simulated cloud cover compared to CALIPSO-GOCCP for most parts of the year, but this negative bias is substantially reduced with the cloud cover increase in A21. The bias reduction with A21 is greatest in autumn and winter, when M92 overestimates the ice cloud fraction but underestimates the liquid cloud fraction by between 0.1 and 0.2. The negative cloud cover bias is largest in spring, and while it is reduced in A21 it still remains at around 0.2. In July, however, the simulations have a smaller positive cloud cover bias of around 0.05. This is due to an overestimation of liquid cloud in both simulations, and in A21 this positive bias increases slightly. It should be noted that the cloud fractions in Fig. 8 are from the satellite simulator COSP (Bodas-Salcedo et al., 2011), not the direct model output shown in Fig. 3a and Fig. B1.

We have also added the following updates to the conclusion on lines 466-468:

This led to a significant improvement in modelled SLW fractions for each cloud isotherm compared to spaceborne lidar observations using CALIOP, and an overall reduction in cloud cover bias for most of the year.

Moreover, we now also include the TOA shortwave CRE in the comparison with observed radiative fluxes, by adding this figure to Fig. 12 along with the following paragraph on lines 411-418:

*In Fig. 12b, we see the cloud effect on outgoing shortwave radiation at the TOA, which is a larger positive number the more clouds reflect solar radiation back to space. Here we see that there is a positive bias in summer and a smaller negative bias in spring. While the negative bias in spring is slightly improved in A21, the positive bias in summer is slightly enhanced. This is most likely connected to the summer cloud cover bias in the model, as also found by Shaw et al. (2022). Since liquid clouds are already overestimated in summer in M92, with its corresponding high primary ice production, reducing the primary ice production in A21 is not able to improve the cloud cover bias nor the bias in shortwave cloud radiative effect. While we do expect Arctic INP concentrations to be higher in the summer, as previously discussed in Sect. 3.1, it seems likely that this is a model issue that requires further improvement of other cloud processes.*

*We add the following updates to the conclusion on lines 472-475:*

*While these changes are quite large, they generally improve the simulated radiative fluxes compared to observations, apart from a bias increase in summer shortwave cloud radiative effect at TOA. This bias increase suggests that the new INP parameterization in A21 may be too low in summer, although the cloud cover bias most likely requires improvement of other cloud processes as well.*

Are the INP measurements adequately representative of the whole Arctic region across the annual cycle? The authors compare their INP parameterization with others from the Arctic region and show consistency, yet a new study by Raif et al. (2024) has shown that INP concentrations may not always be so low. Additionally, there is good evidence that dust from low latitudes is present in the Arctic atmosphere throughout the year, which may strongly influence the spatial variability of INP availability (Shi et al., 2022). In terms of locally produced INP, there will be important sources that are dependent on the seasonal cycle of sea ice (sea spray aerosol), snow coverage (glacial dust sources), and temperature/sunlight (biological sources). I recommend the authors include these caveats to their discussion/conclusions on the representativeness of their parameterization. Perhaps expand lines 390-395 or 210-212.

*Thanks for pointing this out and we fully agree. To further elaborate on the representativeness of our measurements, we have extended lines 232-238 to the following paragraph:*

*Arctic glacial dust with high ice-nucleating ability has also been shown to play an important role in the lower troposphere during summer months (Tobo et al., 2019; Kawai et al., 2023, Tobo et al. 2024), in addition to the variability coming from low latitude dust sources throughout the year (Shi et al., 2022). It should be noted that our measurements are a snapshot in time. The seasonal Arctic INP cycle certainly contains both lower and higher INP concentrations compared to our measurements, both related to the aforementioned local sources as well as transport from lower latitudes. Importantly, our measurements are only at the surface level, while studies from Shi et al. (2022) and Raif et al. (2024) show that long-range transported dust can play an important role at higher altitudes.*

*We have also added the following on line 459:*

*While our measurements are confined to the ground, recent airborne measurements by Raif et al. (2024) have shown the presence of higher ice-nucleating particle concentrations at cloud level, underscoring the importance of investigating the vertical distribution of INPs, something which has also been found in other study areas by e.g. Knopf et al. (2023) and Moore et al. (2024).*

The INP measurements were made using air sampled via an instrument that does not include aerosols smaller than around 500nm in diameter. Does omitting sub 500nm-sized particles bias the INP concentration measurements? Do the authors have a sense of how many more INP would be measured if the whole size distribution was accounted for?

*The fact that our cutoff size is 500 nm does indeed pose the risk that we are missing some smaller INPs. This is something we cannot be sure about. Many of the measurements we compare ours with use filters with a pore size of 200 nm, like the COMBLE measurements from approximately the same location (Geerts et al., 2022). The fact that they were only measuring during cold air outbreaks and have a larger measurement range than us, makes it a bit challenging to compare. However, the fact that our measurements do not show significantly lower INP concentrations than theirs in our common temperature range is consistent with the understanding that most active INPs have a diameter that is 500 nm or larger (e.g., DeMott et al, 2010, Mason et al. 2016), and that we therefore can assume that we do not exclude a large portion of INPs. We have added the measurements of Geerts et al. (2022) to Fig. 2.*

The purely temperature dependent (L6) parameterization presented here is put forward as a new and simplified representation of Arctic INP, yet without any analysis on the origins of the underlying aerosol its use in future studies is limited. The better representative parameterizations are linked to a quantity or property of the aerosol size distribution (e.g., particles larger than 0.5 um or specific components like dust surface area) that are capable of being dependent on variable properties of the atmosphere / aerosol population. With these we are then able to assess changes in INP distributions for future scenarios etc. Do the authors have information on the composition of the aerosol that is acting as an INP? Is this consistent with other studies? Could the authors convert the INP spectra into a quantity related to the aerosol size distribution such as the active site density ns? This would allow a much more thorough comparison with other INP measurements.

*Unfortunately we do not have more detailed information on the composition of the aerosols we measure. As long as we do not see a clear relationship between aerosol surface area or concentration, we believe it is unlikely that we would gain understanding from relating them directly to these quantities – instead we compare our measurements to others that have also seen a similar property in Arctic aerosols, like Li et al. (2022) and Geerts et al (2022).*

*While we attempt to stress in our manuscript that it is important to understand how INP concentrations vary seasonally, spatially and possibly with warming, we also acknowledge that relating INPs to the ambient variables in a consistent way is very difficult. The important work that has been done by Shi et al. (2022) and Kawai (2023), as well as on marine organics INP parameterizations, are important steps, but as a community, we are still lacking an INP parameterization scheme that is able to fully*

*represent INPs in the Arctic. Until that is in place, we might need simpler parameterizations that represent observations directly, without potentially introducing additional sources of error from aerosol transport modelling, in the event that the INP relevant species are even simulated. One might also argue that another important factor for reducing uncertainty in the Arctic climate predictions is increased resolution, and with limited computational resources this would potentially come at the expense of sophisticated aerosol representation, which would then necessitate simpler INP schemes. You make a very important point, but this is our motivation for using a simplified parameterization. We attempted to clarify this also on lines 68-70:*

*Our purpose with observationally constraining Arctic INPs in a simplified manner is to limit bias sources in the simulation, and investigate the potential of simple aerosol-independent parameterizations when computational resources are a limiting factor.*

*However, to investigate climate feedbacks from cloud phase changes, Arctic INPs should somehow relate to the environment. In future work, it would be interesting to study whether this is possible to do in a more simplified way than explicitly relating INPs to aerosol species.*

Minor comments

L46. Dusts from lower latitudes are also readily transported to the Arctic. See Shi et al. (2022) and references therein.

*Thanks for pointing out this omission, the lines 44-47 has been updated with the following:*

*For the Arctic, marine organic aerosol particles from ocean biological activity have been presented as a potentially important source of INPs (DeMott et al., 2016; Wex et al., 2019; Creamean et al., 2019), in addition to mineral dust being transported from lower latitudes (Shi et al., 2022).*

L66. "To our knowledge…" I think this is likely correct, but there are a few studies that have used Arctic-based INP parameterizations in global models, which should be acknowledged for having a similar methodology. Kawai et al. (2023) incorporated an observations-based ice nucleation parameterization of Arctic dust into CAM5 and Shi et al. (2022) modelled INP concentrations in the E3SMv1 model using a parameterization based on Icelandic dust samples.

*In the revised manuscript we acknowledge these important works, and also state the purpose of our study more clearly with the following change on lines 66-70:*

*In this sense, we are complementing work that has been done previously by English et al. (2014), and the Arctic-specific dust parameterizations for global models of Shi et al. (2022) and Kawai et al. (2023). Our purpose with observationally constraining Arctic INPs in a simplified manner is to limit bias sources in the simulation, and investigate the potential of simple aerosol-independent parameterizations when computational resources are a limiting factor.*

L144 and L229. Is total aerosol surface area for particles greater than 0.5 um diameter?

*Originally, we had calculated the total surface area for all counted particles, meaning all aerosols larger than 300 nm. As the referee is perhaps pointing to, including smaller particles in the total aerosol surface area we compare INPs with is meaningless if we do not expect to measure INPs that are smaller than 500 nm. Excluding particles between 300 and 500 nm does however make little difference to the total aerosol surface area, and the correlation with INP freezing temperatures remains unchanged. Regardless, Fig. A1 has been updated using only total surface area of aerosols with diameter ≥ 0.5 um, along with the figure text.*

*This specification is added on lines L148 and L256:*

*total aerosol surface area → total surface area of aerosols with diameter ≥ 0.5 um*

L146. Is this a sensible number?

*This number is very high, and was originally implemented by Shaw et al. (2022) to avoid potential errors when removing the ice number limit in the Arctic (see the supplement of their article). The purpose of the number is to avoid unphysical secondary ice production (SIP). While we have not performed sensitivity tests to the number ourselves, Shaw et al. report that there are limited changes to supercooled liquid water fractions in their sensitivity tests of the number cap.*

*We have updated the description of the adjustments to secondary ice production slightly on lines 162-165:*

*In addition to changing the heterogeneous ice nucleation scheme in CAM6-Nor, we remove the ice number limit for Arctic mixed-phase clouds. To compensate for potential large increase in ice number due to the removal of the ice number limit, secondary ice production is limited to 1000 m−3s−1, following Shaw et al. (2022). Rime-splintering mechanism is the only secondary ice production mechanism in the model, active at temperatures between -8°C and -3°C.*

*While not perhaps completely comparable, a previous study by Zhao and Liu (2021), showed that even when implementing additional SIP mechanisms into CAM6, their maximum zonally averaged SIP rate was 7 kg-1s-1, around five orders of magnitude lower than the ice number cap here.*

L146. What impact will turning off ice detrainment have and why was it turned off?

*In hindsight, this wording was inaccurate and did not precisely reflect the actual changes made in the model. To clarify, GCMs lack comprehensive microphysics in convective clouds, so the phase of hydrometeors formed in convective cores was coded into CAM in a very simplistic manner. Since this went into CAM, it was discovered that freezing in this parameterization initiates at too high temperatures to match the phase from satellite lidar (Tan & Storelvmo, 2016). We hence alter the temperature ramp with the alteration from Tan & Storelvmo (2016), which has recently*

*been verified to improve observational replication specifically in NorESM2 (Hofer et al., 2024).*

*In CAM6-Nor, the phase of cloud particles being detrained from convective updrafts is governed by a linear temperature ramp, where all cloud particles below -35°C are ice, all particles above -5°C are liquid, and the fraction of detrained cloud ice particles decreases linearly as a function of temperature between these two temperatures. Following Hofer et al. (2024), we have adjusted this temperature ramp, which allows us to calculate effects of heterogeneous nucleation with improved confidence that this process is affecting a realistic amount of detrained liquid clouds. In our case, all cloud particles with temperatures above -35°C are detrained as liquid, but can subsequently freeze as they are incorporated into the stratiform cloud microphysics. It should be stressed that this change is applied to both experiments, with and without the new Arctic INP parameterization.*

*We have changed line 165-167, and moved it to Sect 2.2.3:*

*Additionally, the detrainment of cloud particles through convective updrafts is shifted from having the ice phase fraction of detrained particles decreasing linearly with temperature, to having all cloud particles detrained as liquid above -35°C, following Hofer et al. (2024).*

L151. M92 is for primary ice production but the empirical parameterization is for INP. How do you relate the INP to the primary ice production? Also, do you still have Bigg immersion mode freezing and Young contact freezing? Sometimes the Bigg parameterization is used for rain drops rather than cloud droplets – if this is the case perhaps state this to avoid confusion.

*Thanks for pointing this out, we agree that this was not very clear in the original manuscript. Regarding your first question, the primary ice production in the form of a freezing rate is related to the INP parametrization in the following way. In CAM, the model checks if the number of ice crystals is below the calculated INP density at the given temperature, and if so adds the deficit. This is treated as a rate of freezing by dividing by the microphysics timestep – a sub-step of the overall CAM timestep.*

*Regarding your second question, there are two routines in CAM that come from Bigg, concerning both immersion freezing and heterogeneous freezing of rain drops. While the latter is untouched in our simulations, the immersion freezing parameterization is removed wherever the Meyers' parameterization is updated with the Arctic INP immersion mode parameterization (everywhere north of 66.5 °C and between -37 and 0 °C).*

*We have added the following to line 160-161 to clarify which Bigg (1953) parameterization we are talking about:*

*This is done without changing the routine for heterogeneous freezing of rain drops, which also follows from Bigg (1953).*

*We have also changed lines 170-174 to make the overall implementation clearer:*

*In A21, the activated ice number produced by the parameterizations of Meyers et al. (1992) and Bigg (1953) is replaced if the latitude of the atmospheric column exceeds 66.5°N. For these latitudes, the immersion freezing INPs in the temperature range -37°C to 0°C, are determined by the INP observations at Andenes instead. The new parameterization is a temperature-dependent exponential fit of the Andenes INP measurements (see Fig. 2).*

L210. I suggest the authors include low latitude dust sources too.

*Thank you for the good point, we have changed the lines 232-235 to:*

*Arctic glacial dust with high ice-nucleating ability has also been shown to play an important role in the lower troposphere during summer months (Tobo et al., 2019; Kawai et al., 2023; Tobo et al., 2024), in addition to the variability coming from low latitude dust sources (Shi et al., 2022).*

Figure 2. As this is the first time the INP data is being shown please include uncertainty estimates to the scatter points.

*Thank you for pointing this out, this has been updated in Figure 2 (see below) and described on lines 125-126:*

*The uncertainty of the instrument is estimated to be 0.9 °C (David et al., 2019).*

[Figure]

L261. Do the changes in the cloud fraction profile (less/more cloud at different altitudes) have implications for the change in longwave radiation at the surface?

*The LWP and the low-level cloud cover changes are the most pronounced changes, and what we expect to contribute most to the changes in LW CRE. However, the change in cloud altitude may also play a minor role. We now acknowledge this with adding the following sentences on lines 342-343 and 376-378 respectively:*

*This average increase manifests in the vertical as a cloud fraction increase below ~ 760hPa that is somewhat offset by a slight reduction in cloud fraction above (see Fig. B1a).*

*Taking the longwave emissivity dependence into account, the changes in longwave cloud radiative effect at the surface can largely be explained by the changes in CLWP as well, in addition to the changes in cloud cover. Changes in the cloud vertical profile may also play a role.*

L274. What happens to total water path (TWP) and precipitation? Or is it all WBF? I would be surprised if the differences were entirely attributed to this. It would be useful to have all hydrometeor classes accounted for so we can understand where the liquid has come from / gone. Also, is there a reason TWP is never discussed?

*Due to the nature of our model experiments with fixed sea surface temperatures, the evaporation from the ocean remains unchanged between both model experiments. The sea-ice model runs in a simplified manner where surface fluxes are computed without conserving energy. This affects the potential for precipitation changes, which require careful interpretation. In the figures below, we see the total precipitation changes between M92 and A21 in units of mm per month (30 days). While the total average precipitation in the Arctic only sees a minor reduction (-0.5 mm/month annually), we see some precipitation redistribution that results in more visible regional changes. The most marked changes in total precipitation occur over the Norwegian Sea during autumn and winter, corresponding to where we see large increases in total water path (TWP). This is compensated by a small overall increase in the central Arctic, especially in autumn. These changes correspond to relative changes that are between +/- 5%. Figure I below is added to the appendix of the manuscript as Fig. B4, along with Fig. B3 of how the annually averaged total cloud water path (TWP) changes seasonally with the updated parameterization.*

*We add the following sentence on lines 302-305:*

*Indeed, where we see large increases in total cloud water path, particularly in boreal autumn and winter (see Fig. B3) we also see reductions in total precipitation (see Fig. B4), keeping in mind that such changes require careful interpretation as long as sea surface temperatures and evaporation is prescribed.*

[Figure]

**Figure I: The seasonally average change in total precipitation from M92 to A21.**

[Figure]

**Figure II: The spatially averaged change in total precipitation from M92 to A21.**

[Figure]

**Figure III: The spatially averaged change in total cloud water path from M92 to A21.**

*We also discuss how the annually averaged TWP values correspond with different reanalysis products, by adding the following on lins 309-312:*

*The annual mean total cloud water path north of 60°N is 53.0 gm−2 in A21 and 48.0 gm−2 in M92. Both of these values are on the lower end of the range in mean cloud water path (49.5 gm−2 to 82.7 gm−2) estimated by reanalysis products in this region (Gu et al., 2021). It should be noted that measurements of total cloud water path are known to have a large uncertainty at high latitudes (Khanal et al., 2020), particularly due to difficulties in separating precipitation and cloud particles.*

*The calculations behind Fig. B3 and Fig. B4 are introduced on lines 536-539 and lines 540-542, respectively.*

L345 and others. I'm not sure the use of the word drastic (radical; extreme) is an appropriate choice. Please consider changing all instances throughout the manuscript.

*Thank you for your comment, this has been changed throughout the manuscript. Line*

*References:*

*Geerts, B., Giangrande, S. E., McFarquhar, G. M., Xue, L., Abel, S. J., Comstock, J. M., Crewell, S., DeMott, P. J., Ebell, K., Field, P., Hill, T. C. J., Hunzinger, A., Jensen, M. P., Johnson, K. L., Juliano, T. W., Kollias, P., Kosovic, B., Lackner, C., Luke, E., Lüpkes, C., Matthews, A. A., Neggers, R., Ovchinnikov, M., Powers, H., Shupe, M. D., Spengler, T., Swanson, B. E., Tjernström, M., Theisen, A. K., Wales, N. A., Wang, Y., Wendisch, M., & Wu, P. (2022). The COMBLE Campaign: A Study of Marine Boundary Layer Clouds in Arctic Cold-Air Outbreaks. Bulletin of the American Meteorological Society, 103(5), E1371-E1389. https://doi.org/10.1175/BAMS-D-21-0044.1*

*Hofer, S., Hahn, L.C., Shaw, J.K. et al. Realistic representation of mixed-phase clouds increases projected climate warming. Commun Earth Environ 5, 390 (2024). https://doi.org/10.1038/s43247-024-01524-2*

*Kawai, K., Matsui, H., & Tobo, Y. (2023). Dominant role of Arctic dust with high ice nucleating ability in the Arctic lower troposphere. Geophysical Research Letters, 50, e2022GL102470. https://doi.org/10.1029/2022GL102470*

*Li, G., Wieder, J., Pasquier, J. T., Henneberger, J., and Kanji, Z. A.: Predicting atmospheric background number concentration of ice-nucleating particles in the Arctic, Atmos. Chem. Phys., 22, 14441–14454, https://doi.org/10.5194/acp-22-14441-2022, 2022.*

*Mason, R. H., Si, M., Chou, C., Irish, V. E., Dickie, R., Elizondo, P, Wong, R., Brintnell, M., Elsasser, M., Lassar, W. M., Pierce, K. M., Leaitch, W. R., MacDonald, A. M., Platt, A., Toom-Sauntry, D., Sarda-Estève, R., Schiller, C. L., Suski, K. J., Hill, T. C. J., Abbatt, J. P. D., Huffman, J. A., DeMott, P. J., and Bertram, A. K.: Size-resolved measurements*

*of ice-nucleating particles at six locations in North America and one in Europe, Atmos. Chem. Phys., 16, 1637–1651, https://doi.org/10.5194/acp-16-1637-2016, 2016*

*Shi, Y., Liu, X., Wu, M., Zhao, X., Ke, Z., and Brown, H.: Relative importance of high-latitude local and long-range-transported dust for Arctic ice-nucleating particles and impacts on Arctic mixed-phase clouds, Atmos. Chem. Phys., 22, 2909–2935, https://doi.org/10.5194/acp-22-2909-2022, 2022.*

*Tan, I., & Storelvmo, T. (2016). Sensitivity study on the influence of cloud microphysical parameters on mixed-phase cloud thermodynamic phase partitioning in CAM5. Journal of the Atmospheric Sciences, 73(2), 709-728.*

*Zhao, X., & Liu, X. (2021). Global importance of secondary ice production. Geophysical Research Letters, 48, e2021GL092581.*
*https://doi.org/10.1029/2021GL092581*
*DeMott, P. J., et al. (2010). Predicting global atmospheric ice nuclei distributions and their impacts on climate. Proceedings of the National Academy of Sciences, 107(25), 11217-11222.*

**Other changes to the manuscript**

*Throughout the manuscript, the dates describing the averaging period for the model simulation have been changed in the following way to be more accurate: "2007-04-15 to 2010-03-15" → 2007-04-01 to 2010-03-31*

*To make an easier comparison, Fig. 11 and Fig. 12 in the previous manuscript version have been changed from a full timeline to monthly averages over a 3-year period.*

*Line 138: The header of Sect. 2.2.2 has been changed from "Heterogeneous ice nucleation in CAM6 and CAM5" to Adjustments to cloud ice production in CAM6-Nor, to encompass all changes described, not just changes to ice nucleation.*

*Lines 138-167: Sect. 2.2.2 has been restructured slightly to improve the flow and clarity of the model adjustment description. All changes not related to heterogeneous ice nucleation have been moved to the final paragraph starting on line 162.*

*As we added the measurements of Geerts et al. (2022) to Fig. 2, we also added a short comparison between those measurements and ours:*
*Lins 226-229: The INP measurements from Nordmela (Geerts et al., 2022) are also shown as grey crosses in Fig. 2, and exhibit a similar variability as the Andenes measurements, except for a lower concentration of INPs at freezing temperatures above -15◦C and a more extensive measurement temperature range.*
*Lines 244-245: This contrasts them with the measurements from Nordmela, which were targeted towards cold air outbreaks (Geerts et al., 2022).*

*Lines 498-499: Access to CALIPSO-GOCCP-data added as: The CALIPSO-GOCCP data product can be downloaded from https://climserv.ipsl.polytechnique.fr/ cfmip-obs/Calipso_goccp.html.*

*Lines 501-502: Access to NorESM-code added as: The NorESM model code can be accessed at https://github.com/NorESMhub/NorESM.*

---

## Referee Report (RR1)

**Referee report for "Using a region-specific ice-nucleating particle parameterization improves the representation of Arctic clouds in a global climate model" by Astrid Bragstad Gjelsvik and co-authors.**

I am very happy with the actions that the authors have made in response to my comments and suggestions. All my concerns have been addressed with either modifications to the text, or additional analysis. I am very grateful to the authors for putting in additional effort to improve the manuscript. The revised manuscript is a welcome addition towards understanding aerosol impacts on Arctic clouds and climate. I recommend the revised manuscript is accepted for publication.